# EFFICIENT HALLUCINATION DETECTION FOR LLMS USING UNCERTAINTY-AWARE ATTENTION HEADS

## ABSTRACT

Recent progress in large language models (LLMs) has led to systems capable of producing text with remarkable fluency. However, these models are still prone to factual inaccuracies, often referred to as "hallucinations". One strategy to alleviate this issue is uncertainty quantification (UQ), but most existing approaches are computationally intensive or require supervision. In this work, we propose Recurrent Attention-based Uncertainty Quantification (RAUQ), an unsupervised and efficient framework for identifying hallucinations. The method leverages an observation about transformer attention behavior: when incorrect information is generated, certain "uncertainty-aware" attention heads, tend to reduce their focus on preceding tokens. RAUQ automatically detects these attention heads and combines their activation patterns with token-level confidence measures in a recurrent scheme, producing a sequence-level uncertainty estimate in just a single forward pass. Through experiments on twelve tasks spanning question answering, summarization, and translation across four different LLMs, we show that RAUQ consistently outperforms state-of-the-art UQ baselines. Importantly, it does so with minimal cost, less than 1% additional computation. Since it requires neither labeled data nor extensive parameter tuning, RAUQ serves as a lightweight, plug-and-play solution for real-time hallucination detection in white-box LLMs.

## 1 INTRODUCTION

Large language models have become the de facto backbone of modern NLP systems; yet, the impressive fluency of their responses often conceals various inconsistencies known as "hallucinations" (Huang et al., 2025). There are several ways to address hallucinations, such as post-hoc verification using external knowledge bases (Min et al., 2023), incorporating retrieval-augmented generation to ground outputs in factual data (Lewis et al., 2020), or filtering/altering responses based on the uncertainty of a model (Kuhn et al., 2023; Farquhar et al., 2024). The latter approach, based on uncertainty, is the focus of this work.

Uncertainty is a fundamental concept in machine learning, reflecting the fact that we usually lack complete information about the model's predictions or parameters (Gal & Ghahramani, 2016; Houlsby et al., 2011; Hüllermeier & Waegeman, 2021). High predictive uncertainty typically signals a greater likelihood of hallucinations in the model output. Unlike verification methods that rely on external knowledge sources to detect hallucinations, uncertainty quantification (UQ) leverages the model's internal capabilities, thereby mitigating issues related to the completeness of external sources and offering greater versatility. As shown in previous work, uncertainty scores can be used to detect hallucinations that arise due to limitations of LLM parametric knowledge or due to the ambiguity of requests in various generation tasks (Malinin & Gales, 2021; Geng et al., 2024; Baan et al., 2023), including question-answering, machine translation, text summarization, and speech recognition.

UQ for classification and regression tasks is a well-established area spanning decades of research (Zhang et al., 2019; He et al., 2020; Xin et al., 2021; Wang et al., 2022; Vazhentsev et al., 2023; He et al., 2024a). At the same time, UQ for generative tasks has only recently emerged as an active topic and still features open challenges. A crucial difference over classification is that an LLM performs not a single, but multiple conditionally dependent predictions. While recent work has proposed several promising techniques for quantifying predictive uncertainty in generation, e.g. (Kuhn et al., 2023; Farquhar et al., 2024; Duan et al., 2024; Qiu & Miikkulainen, 2024; Lin

et al., 2024b), prior methods have limitations. Namely, information–based scores such as maximum sequence probability (MSP) and token-level entropy are simple and fast, but often underperform on long-form generation tasks (Zhang et al., 2024; Vazhentsev et al., 2025a). Sampling-based scores offer stronger performance but incur large computational overhead (Kuhn et al., 2023; Lin et al., 2024b; Vashurin et al., 2025). Supervised confidence regressors (Azaria & Mitchell, 2023; CH-Wang et al., 2024), i.e., thin supplementary modules trained on supervised annotation, yield accurate scores, but require costly, task–specific annotation and often fail to generalize to out-of-distribution data or across tasks (Vazhentsev et al., 2025a). Thus, despite the recent surge of developments of UQ for LLMs, there is still a lack of an effective, versatile UQ method that (i) avoids the high computational costs associated with sampling-based approaches, and (ii) is robust across tasks and domains.

In this work, we aim to construct such a method. For this purpose, we peek into the attention weights of the transformer and identify patterns that are highly indicative of the presence of hallucinations. Self-attention matrices encode how strongly each newly generated token attends to its immediate context. We empirically observe a systematic drop in the attention weight to the preceding tokens in specific attention heads precisely at positions where the model later proves to be factually incorrect (Figure 1). Based on this finding, we argue that a small number of attention heads capture the behavior of transformer-based LLMs under uncertainty. We propose a method that automatically identifies such "uncertainty-aware" heads inside individual LLM layers and extracts the token-level signal from them. The method recurrently fuses this signal with token probabilities and confidence scores from previously generated tokens, capturing the conditional dependencies across generation steps. Finally, it aggregates token-level scores across the generated sequence and layers. The resulting sequence-level uncertainty score achieves state-of-the-art performance and demonstrates high robustness to the choice of its single hyperparameter. Moreover, since attention weights are readily available at inference time for white-box LLMs, the method requires no additional generation passes and adds almost no computational overhead to response latency.

**Contributions:**

1. **In-depth analysis** of attention-based patterns in LLMs associated with hallucinations, which uncovers what we term "uncertainty-aware" heads, i.e., attention heads whose signals notably correlate with hallucination occurrences.
2. **RAUQ** (Recurrent Attention-based Uncertainty Quantification) – an *unsupervised* UQ method that turns raw attentions and LLM probabilities into reliable uncertainty scores while adding only <1% latency. RAUQ requires *no* task-specific labels or tuning of hyperparameters for a particular LLM, making it an easy plug-and-play for white-box LLMs.
3. **Thorough experimental evaluation** on four LLMs and 12 benchmarks, spanning summarization, translation, and question answering, showing that RAUQ achieves state-of-the-art results over 15 baselines. We also demonstrate the importance of each component within the method and illustrate that each individually could improve other UQ methods.

## 2  RELATED WORK

Several recent studies have proposed attention-based UQ methods for detecting hallucinations in LLM-generated outputs.

Zhang et al. (2023) use attention weights to propagate uncertainty across generation steps by capturing conditional dependencies, helping to mitigate overconfidence from prior hallucinations. However, attention plays a secondary role, with the method mainly relying on probability and entropy.

Yuksekgonul et al. (2024) perform a mechanistic investigation of attention patterns linked to LLM factual errors and propose a supervised UQ method called SAT Probe. They associate hallucinations with weak attention to so-called "constrained" tokens in the prompt – key prompt elements that narrow down the scope of the answer. However, their experiments show that SAT Probe performs only on par with or slightly better than baselines. In a similar vein, Contextualized Sequence Likelihood (Lin et al., 2024a) leverages attention to important tokens in the input context to reweight the contribution of token logits when computing weighted sequence likelihood. Lookback Lens (Chuang et al., 2024) leverages attention maps to construct features for a supervised hallucination detector. The authors hypothesize that hallucinations correlate with less attention paid to the input context. They compute

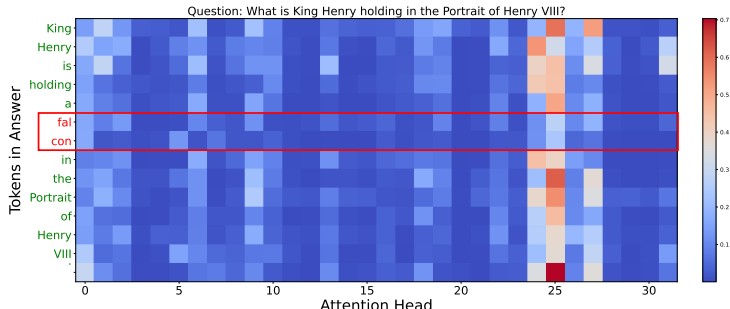

Figure 1: Attention weights in the 29th layer of Llama 3.1 8B from each generated token to its preceding token, given the prompt *What is King Henry holding in the Portrait of Henry VII?*. The $y$ axis specifies the generated tokens, and the $x$ axis specifies the attention heads. Warmer colors indicate higher attention values. The output contains the factually incorrect token *falcon* (the correct answer is *gloves* and *dagger*). Notably, the 25th attention head stands out by consistently assigning relatively high attention to the preceding token. However, for the hallucinated token *falcon*, this attention drops sharply – potentially serving as a signal for hallucination detection.

the ratio between cumulative attention weights to tokens in the answer and the prompt and train a linear classifier on top of these features. Attention-based features are also used in Trainable Attention-Based Dependency (Vazhentsev et al., 2025a). This method adds recurrence when computing uncertainty for subsequent tokens. It demonstrates strong results for in-domain tasks, outperforming Lookback Lens, but both methods lack generalization due to their supervised nature.

Finally, Sriramanan et al. (2024) recently proposed the Attention Score method, where they compute a length-normalized sum of log attention weights to preceding tokens across the prompt and the answer. Lower scores signal the presence of hallucination.

Although recent studies show that attention weights offer valuable signals for detecting hallucinations in LLM outputs, existing methods suffer from various limitations that hinder their effectiveness. SAT Probe, Lookback Lens, and TAD are supervised and show limited generalization beyond their training domain. Zhang et al. (2023) and Lin et al. (2024a) leverage attention only as a supplement to other scores. Sriramanan et al. (2024) do not select proper attention heads before averaging, and allow the attention weights from prompt tokens to participate in the aggregation for the final score, which causes underperformance.

In this work, we aim to overcome the limitations of existing methods. To this end, we identify strong and generalizable attention-based patterns for LLM hallucination detection, isolate the key techniques required to effectively exploit these patterns, and develop a robust *unsupervised* UQ method that achieves state-of-the-art performance.

## 3 HALLUCINATION-ASSOCIATED PATTERNS IN ATTENTION MAPS

We analyze the model's attention maps when an LLM generates correct vs. incorrect outputs. We start with an analysis of attention weights to the immediately preceding token, i.e. $a_{i,i-1}^{lh}$ – attention weight to the $\{i-1\}$-th token during the generation of $i$-th token from the layer $l$ and attention head $h$. Let $N$ be the number of generated tokens in the answer, $H$ the number of attention heads in each layer, and $L$ be the number of layers in the LLM. For illustration, we use the Llama 3.1 8B model.

**Difference between attention weights for hallucinated and non-hallucinated tokens.** Figure 1 presents an example of the attention weights to preceding tokens $a_{i,i-1}^{lh}$ in one of the LLM layers for the input question from the TruthfulQA dataset: *What is King Henry holding in the Portrait of Henry VII?* Most of the generated tokens are aligned with the question. However, the token *falcon* represents a hallucination, i.e. it is factually incorrect (the answer should be *glove and dagger*).

For most attention heads, the weights to previous tokens remain low across all generated tokens. In contrast, the 25th head exhibits a distinct pattern: it assigns relatively high attention to the

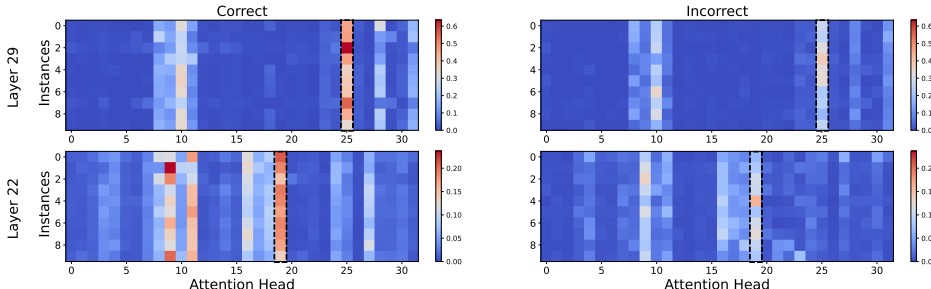

Figure 2: Average attention weights to the preceding token, aggregated over all answer tokens for questions from the TruthfulQA dataset using Llama 3.1 8B. The top 10 highest- and lowest-quality answers, as determined by a quality metric, are labeled as correct and incorrect, respectively. The black dashed box highlights the head with the highest average attention.

preceding token for non-hallucinated (i.e., correct) tokens, but this attention drops significantly for the hallucinated token *falcon*.

This example demonstrates that attention weights from a small subset of attention heads can notably correlate with the factual correctness of generated tokens. While the choice of layer and head might vary, this case suggests that certain heads in specific layers are "uncertainty-aware", i.e., they are sensitive to generation accuracy and could help to identify hallucinations. More examples of the similar pattern for Llama and other LLMs are presented in Figures 6 to 9 in Appendix F.

**Difference between average attention weights for incorrect and correct answers.** We begin by selecting 10 correct and 10 incorrect answers generated by the LLM. To evaluate the correctness of each answer, we use AlignScore – a continuous metric that quantifies semantic similarity between the generated response and the gold-standard answer (Zha et al., 2023). We sort all generations by their AlignScore, and designate the top 10 as correct answers and the bottom 10 as incorrect.

Then, we compute the average attention weight to the previous token across all tokens in the answer using the attention heads in the 29th and 22nd layers of the LLM, i.e. $\bar{a}^{lh} = \frac{1}{N-1} \sum_{i=2}^{N} a_{i,i-1}^{lh}$. Figure 2 presents the resulting values, where each row corresponds to a single selected answer, and each column indicates the average attention weight from a specific head.

The attention maps in the figure demonstrate that certain heads consistently assign higher average attention when the LLM generates correct answers as compared to incorrect ones. Moreover, there is a notable correlation between the quality of the answer and average attention (see Figure 3b). This way, we empirically discovered a pattern for assessing the correctness of LLM generations. From a theoretical perspective, eigenvalue analysis of attention weights reveals similar hallucination patterns, justifying the focus on weights to the previous token, as these are correspond to the log-determinant of the attention matrix (Sriramanan et al., 2024).

**Should we select uncertainty-aware heads, and how should we do it?** We compute the average attention score $\bar{a}^{lh}$ across tokens in two scenarios: (1) attention values are averaged across all heads in a layer, i.e. $\bar{a}^l = \sum_{h=1}^{H} \bar{a}^{lh}$; (2) attention values are extracted from a single head with the *highest* average attention across tokens, i.e. $\bar{a}^{lh_l}$, where $h_l = \arg\max_{h=1...H} \bar{a}^{lh}$. Figure 3a compares the resulting values for correct and incorrect answers.

When using only the selected attention head, we observe a clear difference in the values between correct and incorrect answers. However, averaging attention across all heads eliminates this difference. This once again highlights the importance of focusing on specific uncertainty-aware heads. These heads can be identified by selecting those with the highest average attention weights across all tokens.

**Do we need to look further back at preceding tokens to better detect hallucinations?** We analyze the attention weights to multiple preceding tokens. Here, we compute $a_{i,i-k}^{lh}$ – an attention weight to the $\{i-k\}$-th token ($k$-th preceding token), $k = 1, \ldots, 6$. Figure 4 shows the difference between the average attention weights of the correct and incorrect answers.

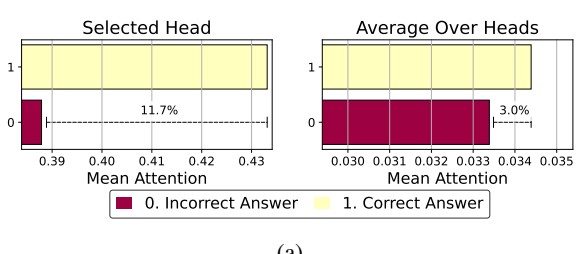 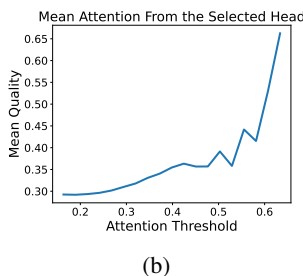

(a)  (b)

Figure 3: Attention weights to the preceding token averaged across all tokens in the generated responses of Llama 3.1 8B on TruthfulQA. a): Comparison between incorrect (AlignScore $< 0.1$) and correct (AlignScore $> 0.9$) answers. Attention values are presented for two scenarios: (left) from the selected head with the highest average attention; (right) averaged across all heads. b): The relationship between average response quality and the average attention weight in the selected head.

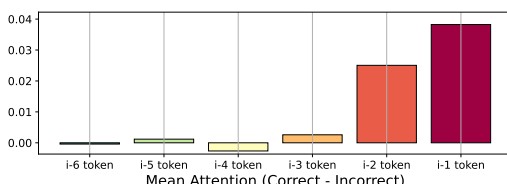

Figure 4: Difference between correct (AlignScore $> 0.9$) and incorrect answers (AlignScore $< 0.1$) in average attention weights to preceding tokens during the generation of answers for the questions from the TruthfulQA dataset using Llama 3.1 8B.

We see that the attention weights differ substantially between correct and incorrect answers only for the two preceding tokens, with almost zero differences observed for earlier tokens. Notably, the difference is substantially larger for the first preceding token as compared to the second one.

**Summary.** Our analysis uncovers attention patterns associated with the factuality of individual tokens and LLM responses in general. A key observation is that such systematic patterns emerge only for a small subset of specific attention heads. Effectively leveraging them requires first identifying the relevant uncertainty-aware attention heads. We also observe that the immediately preceding token provides the strongest signal, leading us to focus solely on it in our method design and subsequent experiments. Below, we leverage the insights from this mechanistic investigation to develop a new *unsupervised* UQ method for LLMs.

## 4 RAUQ: RECURRENT ATTENTION-BASED UNCERTAINTY QUANTIFICATION METHOD

**Key ideas and theoretical grounding.** RAUQ, to be effective, integrates three key ideas.

The first idea is that *attention weights to previous tokens contain patterns indicative for hallucination detection.* This is grounded in previous works on attention-based UQ. Sriramanan et al. (2024) illustrate that attention weights contain patterns indicative of hallucinations through eigen-analysis of attention kernels. They use only the attention weights to the previous token, as these correspond to the eigenvalues of the lower triangular attention matrix, and their sum exactly equals its log determinant. We reveal a similar pattern through a mechanistic analysis of attention weights, examining the correlation between hallucinations and attention weight distributions, as shown in Section 3.

In the second idea, we follow (Zhang et al., 2023; Vazhentsev et al., 2025a) and acknowledge that computing uncertainty at the generation step $i$ requires propagating uncertainty from previous steps due to the conditional dependencies in the probability distribution modeled by the LLM. Namely, even if previous tokens were generated with high uncertainty, a model may condition on them and be highly confident in its current token prediction. To take this issue into account, we introduce a formulation that *recurrently propagates uncertainty from previous steps*.

The third idea is *attention head selection*. We observe that the majority of heads are not indicative of hallucinations (Figure 3a). Therefore, we suggest selecting the most contrastive head that has the best potential for discriminating between hallucinations and non-hallucinations. Our findings are well supported by prior mechanistic interpretability studies of attention heads, which have shown that different heads serve distinct functions Elhelo & Geva (2025).

Let $\mathbf{x}$ be the input sequence and $\mathbf{y} = y_1 y_2 \ldots y_N$ be its corresponding output sequence of length $N$.

**Selecting an attention head in each layer.** For an LLM with $L$ layers and $H$ attention heads per layer, we first select the most informative head. For each layer $l$, we select the head with the maximum average attention weights between consecutive tokens:

$$\mathbf{h}_l(\mathbf{y}) = \arg\max_{h=1\ldots H} \frac{1}{N-1} \sum_{i=2}^{N} a_{i,i-1}^{lh}, \tag{1}$$

where $a_{i,i-1}^{lh}$ is the attention weight from token $y_i$ to $y_{i-1}$ computed by the $h$-th head in layer $l$. By taking the maximum across attention heads within each layer, our method selects the most contrastive attention head that has the best potential for discriminating between hallucinations and non-hallucinations.

**Token-level layer-wise recurrent confidence score.** We recurrently compute the confidence score $\mathbf{c}_l(y_i)$ for the $i$-th token by leveraging the confidence of the previous token $\mathbf{c}_l(y_{i-1})$, the attention weight $a_{i,i-1}^{l\mathbf{h}_l}$ from the selected head $\mathbf{h}_l = \mathbf{h}_l(\mathbf{y})$, and the conditional probability of the current token $P(y_i \mid y_{<i}, \mathbf{x})$ as follows:

$$\mathbf{c}_l(y_i) = \begin{cases} P(y_i \mid \mathbf{x}), & \text{if } i = 1, \\ \alpha \cdot P(y_i \mid y_{<i}, \mathbf{x}) + (1-\alpha) \cdot a_{i,i-1}^{l\,\mathbf{h}_l} \cdot \mathbf{c}_l(y_{i-1}), & \text{if } i > 1, \end{cases} \tag{2}$$

where $\alpha$ is a hyperparameter that balances the contributions of each component. This recurrent formulation also helps to avoid an explosion in confidence scores with an increase in sequence length. We present an ablation study on the impact of varying the parameter $\alpha$ in Section 5.3 and show that a single value provides robust performance across various tasks and even models.

**Sequence-level layer-wise uncertainty score.** Sequence-level errors are typically either (1) *distributed* across all tokens, e.g. in the summarization task; or (2) *localized* in a single fact-related token, e.g. in the QA task. To take into account both cases in the sequence-level uncertainty score, we compute the mean logarithm of the confidence scores across all tokens in the reply (importantly, we do not aggregate scores for tokens in the prompt):

$$\mathbf{u}_l(\mathbf{y}) = -\frac{1}{N} \sum_{i=1}^{N} \log \mathbf{c}_l(y_i). \tag{3}$$

**Final uncertainty score.** Finally, to aggregate the layer-wise uncertainty scores in an unsupervised manner, we compute the maximum uncertainty score across the set of layers:

$$\mathbf{u}(\mathbf{y}) = \max_{l \in \mathcal{L}} \mathbf{u}_l(\mathbf{y}), \tag{4}$$

where $\mathcal{L}$ is a set of the most informative layers. This choice of maximum provides an upper bound on uncertainty. Following previous work (Azaria & Mitchell, 2023; Vazhentsev et al., 2025a), we select these intermediate of the model, as they are the most informative for hallucination detection. An ablation study with various aggregation functions is presented in Section 5.3. The step-by-step description of RAUQ is presented in Algorithm 1.

## 5 EXPERIMENTS

### 5.1 EXPERIMENTAL SETUP

We conducted extensive experiments across three key generation tasks: question answering ("QA"), text summarization ("Summ"), and machine translation ("MT"). We evaluated the effectiveness of UQ in filtering unreliable outputs through selective generation. For all LLMs and tasks, we set $\alpha = 0.2$ and use the same range of layers – from the first third to the second third of the model (e.g., layers 10 to 22 for LLaMA-3.1 8B) without any tuning.

---

**Algorithm 1:** RAUQ: Recurrent Attention-based Uncertainty Quantification method

---

**Data:** Input prompt $\mathbf{x}$, LLM generation $\mathbf{y} = y_{1:N}$, LLM attention weights $a_{i,i-1}^{lh}$ for each layer $l$ and each head $h$, token probabilities $P(y_i \mid y_{<i}, \mathbf{x})$ and a hyperparameter $\alpha$.

**Result:** Uncertainty score $\mathbf{u}(\mathbf{y})$

```
// Selection of uncertainty-aware heads
```
**1 for** $l \leftarrow 1$ **to** $L$ **do**

**2** $\quad \mathbf{h}_l \leftarrow \arg\max_{h=1...H} \frac{1}{N-1} \sum_{i=2}^{N} a_{i,i-1}^{lh}$;

```
// Computing token-level confidence scores with uncertainty-aware heads
```
**3 for** $i \leftarrow 1$ **to** $N$ **do**

**4** $\quad$ **if** $i == 1$ **then**

**5** $\quad\quad$ $\mathbf{c}_l(y_i) \leftarrow P(y_i \mid \mathbf{x})$;

**6** $\quad$ **else**

**7** $\quad\quad$ $\mathbf{c}_l(y_i) \leftarrow \alpha\, P(y_i \mid y_{<i}, \mathbf{x}) + (1-\alpha)\, a_{i,i-1}^{l\mathbf{h}_l}\, \mathbf{c}_l(y_{i-1})$;

```
// Computing layer-wise and final uncertainty scores
```
**8** $\mathbf{u}_l(\mathbf{y}) \leftarrow -\frac{1}{N} \sum_{i=1}^{N} \log \mathbf{c}_l(y_i)$;

**9** $\mathbf{u}(\mathbf{y}) \leftarrow \max_{l \in \mathcal{L}} \mathbf{u}_l(\mathbf{y})$;

**10 return** $\mathbf{u}(\mathbf{y})$;

---

**Datasets.** We consider seven datasets for "QA", three for "Summ", and two for "MT". A detailed description of all datasets is provided in Appendix A, and the dataset statistics are shown in Table 3.

**Models.** To show the generalization of the method across various models, we use several widely used open-weight LLMs: Llama-3.1 8B (Dubey et al., 2024), Qwen-2.5 7B (Yang et al., 2024), Gemma-2 9B (Rivière et al., 2024), and Falcon-3 10B (Falcon-LLM Team, 2024). Additionally, we experiment with open-weight LLMs of diverse sizes: SmolLM-2 360M (Allal et al., 2025), LLaMA-3.2 1B, and LLaMA-3.1 70B (Dubey et al., 2024) in Appendix D.1. Detailed descriptions of generation parameters are presented in Table 3 in Appendix A. We note that different LLMs expose attention weights in various formats. To address this inconsistency, we implement a model-specific normalization procedure that converts raw attention outputs into a unified lower-triangular attention matrix representation across all models.

**Uncertainty quantification baselines.** We compare the proposed RAUQ method with 15 diverse UQ baselines. As a sanity check, we include simple unsupervised baselines such as Maximum Sequence Probability (MSP) and Perplexity (Fomicheva et al., 2020). Among state-of-the-art baselines for whitebox LLMs, we compare our method to Semantic Entropy (Kuhn et al., 2023), hallucination detection with a stronger focus ("Focus") (Zhang et al., 2023), Claim-Conditioned Probability ("CCP") (Fadeeva et al., 2024), EigenScore (Chen et al., 2024), Shifting Attention to Relevance ("SAR") (Duan et al., 2024), Semantic Density (Qiu & Miikkulainen, 2024), and Attention Score (Sriramanan et al., 2024). We also experiment with the "Simple Focus" method, which is a simplified variant of the "Focus" method (Zhang et al., 2023). It preserves only the core scoring components: attention-based signals and greedy log-likelihood, while omitting a proxy model, IDF-based keywords, and NER. Additionally, we consider UQ methods for black-box LLMs, as they also demonstrate strong performance in recent works despite not having access to logits or their hidden states. We use Lexical Similarity based on Rouge-L (Fomicheva et al., 2020), Long-text Uncertainty Quantification ("LUQ") (Zhang et al., 2024), and methods from (Lin et al., 2024b) – Degree Matrix ("DegMat"), Eccentricity, and Sum of Eigenvalues of the graph Laplacian ("EVL").

**Evaluation metrics.** As the main evaluation metric, we use the standard Prediction Rejection Ratio (PRR) (Malinin & Gales, 2021; Vashurin et al., 2025). This is a bounded metric PRR $\in [0; 1]$ that leverages the area under the rejection curve, which plots the average quality of remaining responses when we abstain from a fraction of the most uncertain predictions. We compute PRR over only the first 50% of the curve, as rejecting more than half of the instances is typically impractical. The metric is normalized so that a PRR of zero or below indicates performance at or below the level of random chance, while values approaching one reflect optimal performance. PRR is analogous to ROC-AUC or PR-AUC; however, unlike them, it can be applied not only to discrete quality metrics

Table 1: Mean PRR↑ across tasks for the evaluated LLMs. Warmer color indicates better results.

| UQ Method | Llama-3.1 8B | | | Qwen-2.5 7B | | | Gemma-2 9B | | | Falcon-3 10B | | | Mean |
|---|---|---|---|---|---|---|---|---|---|---|---|---|---|
| | QA | Summ | MT | QA | Summ | MT | QA | Summ | MT | QA | Summ | MT | |
| MSP | .347 | .296 | .397 | .329 | .151 | .369 | .361 | .334 | .381 | .345 | .177 | .333 | .318 |
| Perplexity | .347 | .419 | .380 | .343 | .254 | .406 | .383 | .375 | .405 | .356 | .180 | .439 | .357 |
| CCP | .285 | .307 | .340 | .271 | .186 | .327 | .329 | .345 | .320 | .299 | .128 | .287 | .285 |
| Attention Score | .014 | .126 | .178 | .038 | .130 | .142 | .064 | .103 | .146 | .054 | .192 | .089 | .106 |
| Focus | .320 | .335 | .361 | .264 | .186 | .380 | .416 | .340 | .385 | .313 | .139 | .362 | .317 |
| Simple Focus | .342 | .306 | .415 | .342 | .136 | .399 | .396 | .322 | .422 | .351 | .095 | .385 | .326 |
| DegMat NLI Score entail. | .306 | .118 | .239 | .356 | .154 | .275 | .337 | .138 | .259 | .352 | .132 | .222 | .241 |
| Ecc. NLI Score entail. | .274 | -.008 | .284 | .322 | .002 | .306 | .298 | .020 | .290 | .327 | .038 | .281 | .203 |
| EVL NLI Score entail. | .293 | .114 | .217 | .349 | .154 | .245 | .332 | .133 | .252 | .351 | .135 | .206 | .232 |
| Lexical Similarity Rouge-L | .250 | .131 | .324 | .334 | .131 | .327 | .306 | .161 | .342 | .285 | .084 | .275 | .246 |
| EigenScore | .232 | .078 | .285 | .298 | .061 | .302 | .267 | .106 | .226 | .247 | .051 | .236 | .199 |
| LUQ | .287 | .173 | .214 | .351 | .196 | .213 | .344 | .206 | .259 | .335 | .121 | .196 | .241 |
| Semantic Entropy | .254 | .117 | .315 | .281 | .092 | .317 | .291 | .126 | .337 | .320 | .133 | .291 | .240 |
| SAR | .310 | .170 | .370 | .351 | .153 | .393 | .361 | .235 | .414 | .334 | .094 | .337 | .294 |
| Semantic Density | .330 | .153 | .264 | .352 | .110 | .291 | .375 | .167 | .255 | .358 | .141 | .280 | .256 |
| RAUQ | .396 | .428 | .452 | .358 | .213 | .438 | .421 | .392 | .473 | .392 | .181 | .465 | .384 |

(e.g., correct vs. incorrect answers) but also to continuous ones, such as those commonly used in summarization and MT. For different generation tasks, we use different response quality metrics: accuracy for MMLU and GSM8k; COMET (Rei et al., 2020) for MT; and AlignScore (Zha et al., 2023) for the rest. For summarization tasks, we use AlignScore between the output summary and the input document to measure the factuality of the generation. Additionally, we calculate ROC-AUC using discrete quality metrics obtained by thresholding the original continuous values.

## 5.2 MAIN RESULTS

Table 1 presents the mean PRR for each task (QA, Summ, and MT) for each of the evaluated LLMs. To compute the mean PRR for each task, we average the PRR scores across all relevant datasets, for example, XSum, CNN, and SamSum for summarization. These aggregated PRR scores provide a robust measure of the performance of various methods for each task and model. Detailed results for each model and dataset are presented in Tables 17 to 20 in Appendix E. The results using the ROC-AUC metric are presented in Table 13 in Appendix D.2.

The results demonstrate that the proposed RAUQ method consistently outperforms previous state-of-the-art methods for the QA and translation tasks by a substantial margin across all evaluated LLMs. For instance, for the translation task using Gemma-2 9B, RAUQ largely outperforms the second-best method (Simple Focus) by 0.051 of PRR. In contrast, other single-generation methods based on the attention weights, such as Focus and Attention Score, perform significantly worse.

For summarization, RAUQ also achieves the best results across all models, often with a margin over the second-best method. Notably, RAUQ improves upon the second-best method (MSP) for Gemma-2 9B by 0.017 in terms of PRR. However, for Qwen-2.5 7B in the summarization task, Perplexity achieves the best performance, followed by RAUQ, which outperforms all computationally intensive methods. However, RAUQ consistently outperforms all other sampling-based baselines on average.

Overall, while methods such as MSP, Focus, or SAR might achieve top performance in specific settings, RAUQ demonstrates the most robust performance across all tasks and models, consistently ranking as the best or second-best method by average performance in a task.

Table 12 in Appendix D.1 also provides experimental results with ≤1B and 70B LLMs. These results demonstrate that RAUQ is the best method on average across a wide range of model sizes and tasks, further highlighting its strong generalization ability.

Tables 14 and 15 in Appendix D.3 also provide a comparison with supervised UQ methods. While RAUQ slightly underperforms compared to supervised methods on their in-domain data, it greatly outperforms them on average in out-of-domain scenarios.

## 5.3 Hyperparameter Sensitivity and Ablation Studies

**Impact of the hyperparameter** $\alpha$**.** The hyperparameter $\alpha$ from Equation (2) balances the contributions of attention, confidence from the previous token, and the conditional probability of the current token. When $\alpha$ is equal to 1, RAUQ becomes equivalent to perplexity. When $\alpha$ approaches 0, RAUQ relies solely on the attention weights from the selected head. Figure 5 in Appendix C.1 presents the impact of $\alpha$ on the performance of the RAUQ method for Llama-3.1 8b. For all tasks, except MMLU, the best possible performance is achieved with $\alpha$ between 0.2 and 0.5.

While dataset-specific fine-tuning of $\alpha$ can lead to further improvements, we do not perform such careful tuning in our experiments (Table 1). Instead, we select $\alpha$ using a small out-of-domain subset for Llama-3.1 8b and apply this value uniformly *across all datasets and LLMs*. Despite this, RAUQ achieves consistently strong performance across tasks and LLMs, often achieving the top or near-top results. Strong performance with a fixed hyperparameter underscores the method's robustness.

**Aggregation functions.** Table 5 in Appendix C.1 compares the performance of the RAUQ method using various aggregation functions of token-level confidence scores. We experiment with four aggregation strategies: mean, median, sum of logarithms (inspired by MSP), and mean of logarithms (inspired by perplexity). For the summarization tasks and certain QA datasets such as SciQ, TriviaQA, and GSM8k, mean aggregation yields the best performance. For MMLU, the sum of logarithms substantially outperforms other aggregation strategies, while median aggregation performs second-best for the MedQUAD and TruthfulQA datasets. Overall, however, the top two performing methods are those that apply length normalization. Among them, the mean of logarithms of token-level confidence scores used in RAUQ consistently delivers the strongest results across datasets.

Table 6 in Appendix C.1 compares the performance of RAUQ using various aggregation functions of layer-wise uncertainty scores. We consider three aggregation strategies: mean, median, and maximum. Both maximum and median yield similarly strong performance, while the mean aggregation performs slightly worse. Given that the maximum is a more intuitive choice – it effectively captures the peak uncertainty within a generation and achieves better results in 6 out of 12 tasks, with a slight average improvement of 0.001 PRR across tasks over the median, we adopt it as the default layer-wise aggregation method in our experiments.

**Recurrent uncertainty propagation functions.** Table 7 in Appendix C.1 presents the performance of the RAUQ method using various recurrent formulas for the calculation of token-level confidence scores. We consider five modifications of Equation (2): (1) removing attention weights, (2) removing recurrence, (3) replacing the confidence score of the previous token with its probability, (4) multiplying probabilities with attentions, and (5) the recurrent formula proposed in RAUQ.

The proposed formula achieves the best results on the majority of the datasets. Removing either recurrence or attention often leads to substantially worse performance. The results highlight the importance of each component in the proposed formula for achieving good results.

**Layers and heads selection.** Table 8 in Appendix C.2 shows RAUQ performance across various layer subsets. The results indicate that using a subset of middle layers consistently achieves strong performance, while selecting an optimal single layer offers only marginal improvements and requires supervision. Tables 9 and 10 in Appendix C.2 presents selected attention heads for WMT14 De-En and CoQA. They show that the most informative heads are highly consistent within tasks and largely overlapping across tasks, emphasizing both intra-task and cross-task stability of the RAUQ method. Table 11 in Appendix C.2 shows results when a single optimal head per layer is selected on a small validation set. The average performance across all datasets remains similar, which indicates that our dynamic, fully unsupervised strategy already achieves near-optimal performance without task-specific tuning, preserving its plug-and-play nature.

**Alternative interpretability scores.** Table 16 in Appendix D.4 shows RAUQ performance when Layer Integrated Gradients (LIG) (Sundararajan et al., 2017) are used in place of attention scores. We replace the original attention weights with LIG scores computed on the output projection layer and partitioned to match the original multi-head structure. The results show only a 0.4% average drop in PRR, confirming that RAUQ does not critically depend on standard attention mechanisms and can be extended to models with non-standard or without attention mechanisms.

Table 2: PRR↑ for Llama-3.1 8B across various modifications of the Attention Score method incorporating components from RAUQ. The best method is in **bold**, the second best is underlined.

| UQ Method | XSum | SamSum | CNN | WMT14 | WMT19 | MedQUAD | TruthfulQA | CoQA | SciQ | TriviaQA | MMLU | GSM8k | Mean |
|---|---|---|---|---|---|---|---|---|---|---|---|---|---|
| Attention Score | .036 | .083 | .258 | .176 | .179 | -.295 | .081 | -.028 | -.142 | .067 | .209 | .209 | .069 |
| Attention Score (Gen. Tokens) | .020 | .117 | .261 | .196 | .198 | -.305 | -.020 | .064 | .124 | .130 | .232 | .192 | .101 |
| Attention Score (Gen. Tokens, Selected Head) | .154 | -.043 | .351 | .187 | .200 | -.113 | -.025 | .092 | .161 | .151 | .414 | .197 | .144 |
| RAUQ | **.370** | **.464** | **.452** | **.394** | **.509** | **.241** | **.364** | **.265** | **.506** | **.522** | **.549** | **.323** | **.413** |

**Extending our findings to the Attention Score method.** To demonstrate the robustness and generalization of RAUQ components, we integrated them into the recently published Attention Score (AS) method (Sriramanan et al., 2024), resulting in two modifications. We compare (1) the original official implementation of AS; (2) AS that uses only the attention weights of the generated tokens, excluding the prompt; (3) AS that combines the previous feature and implements also the selection of the uncertainty-aware attention heads; (4) the full RAUQ method with recurrence.

Results in Table 2 show that excluding contributions from prompt tokens significantly improves Attention Score, yielding a 0.032 improvement in PRR. The highest improvement is achieved on SciQ, CoQA, and TriviaQA. Incorporating attention head selection further boosts the average performance by 0.043, with a large gain of 0.182 on MMLU. Nevertheless, our full method further incorporates token probabilities and recurrently aggregates uncertainty scores from previous generation steps, which provides a distinct advantage. Overall, these results suggest that our findings regarding attention heads and design choices in RAUQ are systematic and generalize to prior UQ methods as well.

**Qualitative analysis.** We analyzed samples with the highest and lowest RAUQ scores for LLaMA-3.1 8B on the TruthfulQA dataset. RAUQ effectively detects erroneous generations, with most of the detected errors attributed as factual and reasoning errors. Most of the erroneous generations with low uncertainty correspond to common misbeliefs. The detailed results are presented in Table 21 in Appendix G.

### 5.4 COMPUTATIONAL EFFICIENCY

To demonstrate the computational efficiency of RAUQ, we conducted a comprehensive runtime comparison against other state-of-the-art UQ methods using Llama-3.1 8b. All experiments were performed on a single 80GB NVIDIA H100 GPU using single-batch inference, following the same setup as in Table 1. Table 4 in Appendix B reports the average runtime per instance for each UQ method, and quantifies their computational overhead relative to standard LLM inference without UQ.

State-of-the-art UQ methods such as DegMat (Farquhar et al., 2024), Semantic Entropy (Kuhn et al., 2023), and SAR (Zhang et al., 2023) introduce huge computational overhead (400–800%) due to repeated sampling from the LLM. In contrast, the RAUQ method introduces less than 1% overhead since it does not require sampling or inference of an auxiliary model, making it a fast, lightweight, and plug-and-play solution for any white-box LLM.

## 6 CONCLUSION

We introduced RAUQ, an unsupervised, attention-based framework that converts the intrinsic signals already produced by every transformer layer into reliable sequence-level uncertainty scores with a single forward pass. A simple head-selection heuristic, a recurrent confidence propagation rule, and a length-normalized aggregation allow RAUQ to capture both local spikes and global drifts in confidence without external supervision or multiple sampling. Extensive experiments on 12 datasets spanning question answering, abstractive summarization, and machine translation, and on four open-weight LLMs show that RAUQ delivers state-of-the-art performance. Moreover, RAUQ adds only <1 % latency overhead, making it a practical off-the-shelf UQ technique.

### ETHICAL STATEMENT

In this work, we propose RAUQ, a plug-and-play method for real-time hallucination detection in white-box LLMs that requires no task-specific labels or multiple samples. RAUQ is efficient, easy to integrate, and demonstrates significant performance improvements over baseline methods in our

experiments. We believe that our work is a meaningful step toward more trustworthy and responsible use of LLMs, particularly in safety-critical domains such as healthcare and legal documentation. In our experiments, we considered open-source LLMs and datasets not aimed at harmful content. Furthermore, our approach poses no negative social impact, as it does not rely on sensitive data, user annotations, or other elements that might raise ethical concerns. Finally, RAUQ uses raw attention weights without any processing, and thus may reflect biases inherent in the underlying model. However, it does not amplify them, as it involves no modification to the model or additional parameters.

We used writing assistants when working on this paper, in order to improve grammatical accuracy.

## REPRODUCIBILITY STATEMENT

The full codebase, including configuration files and scripts for reproducing the experiments, is provided as supplementary material. Additionally, details of the generation hyperparameters and dataset statistics are presented in the Appendix A

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

# A DATASET AND GENERATION STATISTICS

For QA, we use seven datasets: TruthfulQA (Lin et al., 2022) – a benchmark for assessing the truthfulness of LLM responses, SciQ (Welbl et al., 2017) for scientific QA, MMLU (Hendrycks et al., 2021) – a standard multitask evaluation benchmark, TriviaQA (Joshi et al., 2017) for trivia questions, CoQA (Reddy et al., 2019) for conversational QA, MedQUAD (Ben Abacha & Demner-Fushman, 2019) for medical questions, and GSM8k (Cobbe et al., 2021) for mathematical reasoning. For summarization, we use three datasets with different summarization types: CNN/DailyMail (See et al., 2017) for news article summarization, SamSum (Gliwa et al., 2019) for dialogue summarization, and XSum (Narayan et al., 2018) for summarizing into a single sentence. For the MT task, we evaluate on two language pairs from WMT: German–English from WMT19 (Barrault et al., 2019) and French–English from WMT14 (Bojar et al., 2014).

The detailed description of the used datasets and the generation parameters of LLMs is presented in Table 3. For all LLMs, we used the same generation hyperparameters, while for each dataset, we separately fixed the number of few-shot and maximum generation length. We use greedy decoding to generate the main sequence, for which we compute uncertainty, while sampling is used solely to obtain multiple outputs for sampling-based baselines. Accordingly, the MSP score is always computed on the greedy output sequence (Aichberger et al., 2024).

Table 3: Statistics of the datasets and generation parameters of the used LLMs. For all datasets, we do not limit the maximum input length.

| Task | Dataset | Number of test samples | N-shot | Generation length | Do sample | Temperature | Top-p | Beams | Repetition Penalty |
|------|---------|------------------------|--------|-------------------|-----------|-------------|-------|-------|--------------------|
| QA | TruthfulQA | 817 | 5 | 128 | False | 1.0 | 1.0 | 1 | 1 |
| | SciQ | 1000 | 0 | 20 | | | | | |
| | MMLU | 2000 | 5 | 3 | | | | | |
| | TriviaQA | 2000 | 5 | 20 | | | | | |
| | CoQA | 2000 | all preceding questions | 20 | | | | | |
| | MedQUAD | 1000 | 5 | 128 | | | | | |
| | GSM8k | 1319 | 5 | 256 | | | | | |
| ATS | CNN/DailyMail | 2000 | 0 | 128 | | | | | |
| | SamSum | 819 | 0 | 128 | | | | | |
| | XSum | 2000 | 0 | 128 | | | | | |
| NMT | WMT19 (DE-EN) | 2000 | 0 | 107 | | | | | |
| | WMT14 (FR-EN) | 2000 | 0 | 107 | | | | | |

# B COMPUTATIONAL EFFICIENCY

Table 4: Inference runtime of UQ methods measured on all test instances from all datasets with generations from Llama-3.1 8b. The best results are in **bold**.

| UQ Method | Runtime per batch | Overhead |
|-----------|-------------------|----------|
| MSP | 1.16±0.45 | - |
| DegMat NLI Score Entail. | 6.40±1.76 | 450% |
| Lexical Similarity ROUGE-L | 6.11±1.75 | 425% |
| Semantic Entropy | 6.40±1.76 | 450% |
| SAR | 10.71±3.21 | 820% |
| Semantic Density | 6.27±1.76 | 438% |
| RAUQ | 1.17±0.45 | **0.3%** |

## C   RESULTS OF ABLATION STUDIES

### C.1   AGGREGATION STRATEGIES AND HYPERPARAMETER SENSITIVITY

Tables 5 to 7 present the performance of the RAUQ method using various aggregation functions for token-level confidence scores, layer-wise uncertainty scores, and various recurrent formulas for computing token-level confidence scores, respectively. Figure 5 shows the impact of $\alpha$ on the performance of the RAUQ method for Llama-3.1 8B.

Table 5: PRR↑ for Llama-3.1 8b model for various aggregation function of token-level confidence scores. The best method is in **bold**, the second best is underlined.

| Token Aggregation | XSum | SamSum | CNN | WMT14 | WMT19 | MedQUAD | TruthfulQA | CoQA | SciQ | TriviaQA | MMLU | GSM8k | Mean |
|---|---|---|---|---|---|---|---|---|---|---|---|---|---|
| $-\frac{1}{N}\sum_{i=1}^N \mathbf{c}_l(t_i)$ | **.375** | .419 | **.460** | .359 | .485 | .140 | .304 | .259 | **.511** | **.534** | .526 | **.339** | .393 |
| $-\text{median}_{i=1}^N \mathbf{c}_l^t(t_i)$ | .267 | .403 | .437 | .249 | .340 | .154 | .317 | .234 | .430 | .432 | .635 | .253 | .346 |
| $-\sum_{i=1}^N \log \mathbf{c}_l^t(t_i)$ | .027 | -.045 | .325 | .224 | .242 | .107 | .035 | .114 | .202 | .300 | **.658** | .213 | .198 |
| $-\frac{1}{N}\sum_{i=1}^N \log \mathbf{c}_l^t(t_i)$ | .370 | **.464** | .452 | **.394** | **.509** | **.249** | **.364** | **.265** | .506 | .522 | .549 | .323 | **.413** |

Table 6: PRR↑ for Llama-3.1 8b model for various aggregation function of layer-wise uncertainty scores. The best method is in **bold**, the second best is underlined.

| Layer Aggregation | XSum | SamSum | CNN | WMT14 | WMT19 | MedQUAD | TruthfulQA | CoQA | SciQ | TriviaQA | MMLU | GSM8k | Mean |
|---|---|---|---|---|---|---|---|---|---|---|---|---|---|
| $\frac{1}{|\mathcal{L}|}\sum_{l\in\mathcal{L}} \mathbf{u}_l(y)$ | **.384** | .419 | **475** | .389 | .519 | .154 | .345 | **.274** | .496 | **.535** | .529 | .337 | .404 |
| $\text{median}_{l\in\mathcal{L}} \mathbf{u}_l(y)$ | .378 | .426 | .471 | .388 | **.526** | .246 | .351 | .267 | .502 | .532 | .532 | **.340** | .412 |
| $\max_{l\in\mathcal{L}} \mathbf{u}_l(y)$ | .370 | **.464** | .452 | **.394** | .509 | **.249** | **.364** | .265 | **.506** | .522 | **.549** | .323 | **.413** |

Table 7: PRR↑ for Llama-3.1 8b model for various function for recurrent calculation of confidence scores $\mathbf{c}_l(t_i)$ in Equation (2). The best method is in **bold**, the second best is underlined.

| Recurrent Formula | XSum | SamSum | CNN | WMT14 | WMT19 | MedQUAD | TruthfulQA | CoQA | SciQ | TriviaQA | MMLU | GSM8k | Mean |
|---|---|---|---|---|---|---|---|---|---|---|---|---|---|
| $\alpha \cdot P(t_i \mid \mathbf{x}, t_{<i}) + (1-\alpha) \cdot \mathbf{c}_l(t_{i-1})$ | .330 | .383 | .393 | .238 | .313 | **.273** | .224 | .267 | .273 | .514 | .475 | .279 | .330 |
| $\alpha \cdot P(t_i \mid \mathbf{x}, t_{<i}) + (1-\alpha) \cdot a_{i,i-1}^{l, \mathbf{h}_i}$ | **.412** | .387 | .457 | .332 | .436 | .205 | .322 | .257 | .485 | .517 | .550 | .305 | .388 |
| $\alpha \cdot P(t_i \mid \mathbf{x}, t_{<i}) + (1-\alpha) \cdot a_{i,i-1}^{l, \mathbf{h}_i} \cdot P(t_{i-1} \mid \mathbf{x}, t_{<i-1})$ | .399 | .421 | **.461** | .370 | .472 | .235 | .336 | **.279** | .456 | .517 | .532 | .318 | .399 |
| $P(t_i \mid \mathbf{x}, t_{<i}) \cdot a_{i,i-1}^{l, \mathbf{h}_i}$ | .394 | .327 | .459 | .226 | .337 | .149 | .251 | .161 | .330 | .330 | **.645** | .255 | .322 |
| $\alpha \cdot P(t_i \mid \mathbf{x}, t_{<i}) + (1-\alpha) \cdot a_{i,i-1}^{l, \mathbf{h}_i} \cdot \mathbf{c}_l(t_{i-1})$ | .370 | **.464** | .452 | **.394** | **.509** | .249 | **.364** | .265 | **.506** | **.522** | .549 | **.323** | **.413** |

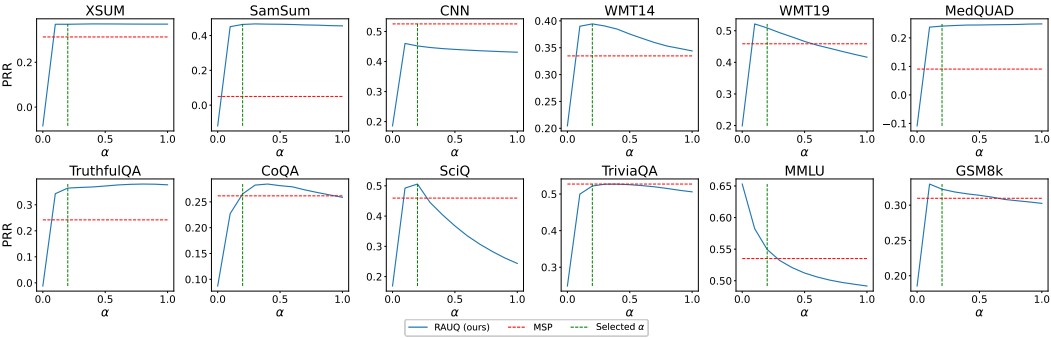

Figure 5: PRR↑ as a function of the hyperparameter $\alpha$ for Llama-3.1 8B. The vertical line marks the value of $\alpha$ used in our experiments.

### C.2   LAYERS AND HEADS SELECTION

**Layer selection analysis.** Table 8 present the performance of the RAUQ method across various layer subsets. We compare RAUQ using individual layers, all layers, and aggregated middle layers. In our experiments, we consistently use the same range of layers – from the first third to the second third of the model (e.g., layers 10 to 22 for LLaMA-3.1 8B) without any task- or model-specific tuning.

The results indicate that although certain layers (e.g., the 25th or 30th) perform better on specific tasks, they tend to underperform on average. While the selection of the optimal layer (e.g., 22nd for LLaMA-3.1 8B) can slightly improve overall performance, it requires supervision, whereas our

Table 8: PRR↑ for Llama-3.1 8b model for various layer subsets $\mathcal{L}$ in Equation (4). The best method is in **bold**, the second best is underlined.

| Layer Subset | WMT14 | WMT19 | MedQUAD | TruthfulQA | CoQA | SciQ | Mean |
|---|---|---|---|---|---|---|---|
| RAUQ (22nd layer) | **.412** | **.529** | .237 | .354 | .267 | **.514** | **.385** |
| RAUQ (25th layer) | .359 | .519 | .244 | **.382** | .262 | .462 | .371 |
| RAUQ (30th layer) | .272 | .433 | .168 | .326 | **.268** | .456 | .320 |
| RAUQ (5 middle layers, 14–18) | .388 | .502 | .240 | .365 | .258 | .510 | .377 |
| RAUQ (All layers) | .386 | .516 | **.244** | .366 | .260 | .490 | .377 |
| RAUQ | .394 | .509 | .241 | .364 | .265 | .506 | .380 |

method is designed to be fully unsupervised. Using all layers instead of only the middle layers yields only a marginal decrease in performance for RAUQ on LLaMA-3.1 8B, with an average drop of just 0.003 in PRR. Therefore, while this modification can slightly enhance results, it is not a critical component of our method.

**Head selection analysis.** Tables 9 and 10 present an analysis of the selected attention heads for the WMT14 De-En and CoQA datasets using LLaMA-3.1 8B. We report the top-3 heads based on their selection frequency according to our criterion, along with the corresponding percentages.

First, the tables show that in most cases, the most frequently selected head accounts for over 90% of instances, indicating high stability in head selection. Even in layers where head selection is less consistent, the top-3 heads still cover more than 90% of cases, suggesting that the model typically chooses similar heads across inputs within the same task.

Second, when comparing selected heads across the two datasets, we observe substantial overlap. For example, in layers 10, 12, 13, 15, 16, and 20, the selected heads are fully aligned, reflecting strong cross-task consistency. Overall, while some variation exists, the same heads generally provide the most informative signals used in the RAUQ method, highlighting both intra-task and cross-task stability.

Table 9: The top three most frequently selected attention heads per layer in the Llama-3.1 8B model on the WMT14 dataset with its selection frequency according to our criterion.

| Attention Head | Layer 10 | Layer 11 | Layer 12 | Layer 13 | Layer 14 | Layer 15 | Layer 16 | Layer 17 | Layer 18 | Layer 19 | Layer 20 | Layer 21 | Layer 22 |
|---|---|---|---|---|---|---|---|---|---|---|---|---|---|
| Top-1 head | 10 (87.5%) | 10 (99.2%) | 12 (100.0%) | 28 (100.0%) | 19 (84.2%) | 6 (99.9%) | 30 (99.7%) | 12 (83.0%) | 29 (46.2%) | 11 (97.2%) | 3 (99.7%) | 10 (50.9%) | 9 (99.3%) |
| Top-2 head | 0 (12.3%) | 16 (0.6%) | - | - | 14 (12.9%) | 24 (0.1%) | 22 (0.4%) | 22 (16.4%) | 14 (29.0%) | 8 (2.2%) | 0 (0.3%) | 9 (26.7%) | 19 (0.3%) |
| Top-3 head | 18 (0.1%) | 12 (0.1%) | - | - | 8 (2.5%) | - | - | 6 (0.3%) | 26 (11.1%) | 10 (0.5%) | - | 3 (15.4%) | 11 (0.2%) |

Table 10: The top three most frequently selected attention heads per layer in the Llama-3.1 8B model on the CoQA dataset with its selection frequency according to our criterion.

| Attention Head | Layer 10 | Layer 11 | Layer 12 | Layer 13 | Layer 14 | Layer 15 | Layer 16 | Layer 17 | Layer 18 | Layer 19 | Layer 20 | Layer 21 | Layer 22 |
|---|---|---|---|---|---|---|---|---|---|---|---|---|---|
| Top-1 head | 0 (95.2%) | 10 (76.8%) | 12 (100.0%) | 28 (100.0%) | 16 (27.0%) | 6 (95.0%) | 30 (91.3%) | 12 (76.7%) | 29 (54.6%) | 11 (61.5%) | 3 (66.0%) | 10 (74.7%) | 9 (64.2%) |
| Top-2 head | 10 (4.3%) | 23 (7.3%) | - | - | 8 (26.5%) | 24 (2.8%) | 22 (8.6%) | 6 (18.9%) | 25 (17.5%) | 8 (17.5%) | 0 (23.8%) | 8 (7.8%) | 19 (26.5%) |
| Top-3 head | 18 (0.4%) | 31 (5.1%) | - | - | 14 (17.1%) | 4 (0.7%) | 9 (0.1%) | 30 (1.2%) | 26 (15.5%) | 23 (3.3%) | 27 (3.8%) | 9 (5.9%) | 18 (2.4%) |

**Experiments with a single optimal head.** Table 11 presents an analysis in which a single optimal head per layer is selected for all inputs determined on a small held-out validation set of 100 instances per task. The results show that the gains from such precise per-dataset head selection are marginal, and the average performance across all datasets remains effectively similar. This indicates that retaining dynamic, unsupervised head selection as part of the algorithm fully removes the need for any precise task-specific adjustments and already achieves near-optimal performance. This design choice also ensures that the method remains entirely unsupervised, requires no validation data, and is seamlessly plug-and-play for any new LLM or task.

Table 11: PRR↑ for Llama-3.1 8b model for RAUQ with dynamic head selection per input and with a single optimal head per layer, fixed across all inputs. The best method is in **bold**.

| UQ Method | XSum | SamSum | CNN | WMT14 | WMT19 | MedQUAD | TruthfulQA | CoQA | SciQ | TriviaQA | MMLU | GSM8k | Mean |
|---|---|---|---|---|---|---|---|---|---|---|---|---|---|
| RAUQ | **.384** | .423 | .189 | .406 | **.488** | **.317** | **.399** | .248 | **.506** | **.548** | .513 | .323 | **.395** |
| RAUQ (Single Head) | .382 | **.426** | **.195** | **.407** | .481 | .303 | .386 | **.257** | .494 | .544 | **.528** | **.325** | .394 |

# D  ADDITIONAL EXPERIMENTAL RESULTS

## D.1  EXPERIMENTS WITH DIVERSE LLM SIZES

To demonstrate that RAUQ generalizes effectively to both larger and smaller LLMs, we have conducted additional experiments using SmolLM-2 360M, LLaMA-3.2 1B, and LLaMA-3.1 70B. The results are presented in Table 12. For models with $\leq$1B parameters, we exclude MMLU, GSM8K, and MedQUAD due to their near-zero performance on these tasks.

The results show that RAUQ is the best method for QA and MT on $\leq$1B LLMs, and for MT on the 70B LLM, while it is the second-best for QA on the 70B LLM. Overall, RAUQ surpasses the second-best method by an average of 2% of PRR across all tasks and models. These results highlight the strong generalization ability of RAUQ across a wide range of model sizes.

Table 12: Mean PRR$\uparrow$ across tasks for the evaluated LLMs ($\leq$1B and 70B). Warmer color indicates better results.

| UQ Method | SmolLM-2 360M | | | Llama-3.2 1B | | | Llama-3.2 70B | | | Mean |
| --- | --- | --- | --- | --- | --- | --- | --- | --- | --- | --- |
| | QA | Summ | MT | QA | Summ | MT | QA | Summ | MT | |
| MSP | .360 | .449 | .330 | .324 | .507 | .351 | .364 | .128 | .447 | .362 |
| Perplexity | .371 | .330 | .487 | .310 | .392 | .427 | .323 | .245 | .335 | .358 |
| CCP | .281 | .457 | .361 | .283 | .517 | .328 | .350 | .135 | .387 | .344 |
| Attention Score | .071 | .004 | .120 | .051 | .033 | .103 | .053 | .045 | .213 | .077 |
| Simple Focus | .401 | .429 | .410 | .370 | .488 | .424 | .380 | .128 | .435 | .385 |
| DegMat NLI Score entail. | .342 | .059 | .227 | .305 | .078 | .287 | .380 | .091 | .273 | .227 |
| Ecc. NLI Score entail. | .209 | -.013 | .169 | .225 | -.012 | .293 | .330 | -.003 | .298 | .166 |
| EVL NLI Score entail. | .333 | .055 | .216 | .298 | .072 | .268 | .369 | .091 | .265 | .219 |
| Lexical Similarity Rouge-L | .290 | -.013 | .193 | .255 | .074 | .337 | .362 | .089 | .332 | .213 |
| EigenScore | .173 | .068 | .061 | .145 | .029 | .301 | .296 | .044 | .325 | .160 |
| LUQ | .337 | .076 | .242 | .279 | .118 | .263 | .376 | .139 | .254 | .232 |
| Semantic Entropy | .201 | .067 | .227 | .187 | .084 | .268 | .309 | .069 | .373 | .198 |
| SAR | .343 | .095 | .348 | .295 | .091 | .408 | .382 | .106 | .372 | .271 |
| Semantic Density | .357 | .209 | .259 | .348 | .217 | .285 | .385 | .100 | .239 | .267 |
| RAUQ | .425 | .356 | .490 | .356 | .423 | .495 | .360 | .245 | .457 | .401 |

## D.2  EXPERIMENTS USING THE ROC-AUC METRIC

The results evaluated using the ROC-AUC metric are presented in Table 13. For all generation quality metrics except accuracy, we compute scores by thresholding the original continuous values to obtain discrete versions of the quality metrics. The thresholds were empirically determined as follows: 0.5 for QA and Summ, and 0.85 for MT.

We observe similar trends to those with the PRR metric. RAUQ significantly outperforms all other methods for summarization and MT tasks. For QA, RAUQ is the best method for Llama-3.1 8B and Falcon-3 10B, while performing comparably to computationally intensive sampling-based approaches for other models. Overall, RAUQ achieves a 0.6% improvement over the second-best method (Perplexity) across all evaluated models.

## D.3  COMPARISON WITH SUPERVISED METHODS

We compare our method against several state-of-the-art supervised methods that leverage hidden states or attention weights: Factoscope (He et al., 2024b), SAPLMA (Azaria & Mitchell, 2023), MIND (Su et al., 2024), Sheeps (CH-Wang et al., 2024), LookBack Lens (Chuang et al., 2024), SATRMD+MSP (Vazhentsev et al., 2025b), and TAD (Vazhentsev et al., 2025a). We evaluate these methods in two scenarios: in-domain, where the model is trained directly on the target task, and out-of-domain, where the model is trained on all datasets except one, which is held out for testing. Tables 14 and 15 show the performance of supervised methods in the in-domain and out-of-domain settings respectively.

Table 13: Mean ROC-AUC↑ across tasks for the evaluated LLMs. Warmer color indicates better results.

| UQ Method | Llama-3.1 8B | | | Qwen-2.5 7B | | | Gemma-2 9B | | | Falcon-3 10B | | | Mean |
|---|---|---|---|---|---|---|---|---|---|---|---|---|---|
| | QA | Summ | MT | QA | Summ | MT | QA | Summ | MT | QA | Summ | MT | |
| MSP | .711 | .718 | .686 | .700 | .559 | .685 | .746 | .735 | .683 | .721 | .583 | .688 | .685 |
| Perplexity | .701 | .812 | .690 | .705 | **.661** | .713 | .735 | .766 | .699 | .713 | **.606** | .715 | .710 |
| CCP | .685 | .705 | .648 | .668 | .575 | .658 | .729 | .731 | .646 | .703 | .569 | .657 | .665 |
| Attention Score | .497 | .552 | .553 | .522 | .530 | .540 | .519 | .536 | .543 | .534 | .590 | .539 | .538 |
| Focus | .698 | .746 | .663 | .642 | .612 | .682 | .747 | .738 | .684 | .699 | .577 | .672 | .680 |
| Simple Focus | .718 | .730 | .694 | .703 | .588 | .700 | **.753** | .723 | .706 | .724 | .543 | .691 | .689 |
| DegMat NLI Score entail. | .676 | .591 | .618 | .691 | .604 | .637 | .692 | .612 | .636 | .700 | .581 | .620 | .638 |
| Ecc. NLI Score entail. | .659 | .498 | .630 | .682 | .510 | .650 | .678 | .535 | .642 | .688 | .546 | .648 | .614 |
| EVL NLI Score entail. | .668 | .590 | .610 | .688 | .602 | .630 | .690 | .607 | .632 | .703 | .583 | .612 | .635 |
| Lexical Similarity Rouge-L | .659 | .605 | .660 | .687 | .594 | .677 | .684 | .620 | .668 | .673 | .559 | .646 | .644 |
| EigenScore | .643 | .533 | .629 | .675 | .549 | .655 | .658 | .592 | .614 | .662 | .527 | .623 | .613 |
| LUQ | .667 | .633 | .618 | .688 | .627 | .613 | .690 | .644 | .629 | .687 | .570 | .599 | .639 |
| Semantic Entropy | .661 | .583 | .658 | .680 | .544 | .665 | .683 | .595 | .661 | .706 | .579 | .666 | .640 |
| SAR | .696 | .627 | .692 | **.708** | .590 | .710 | .723 | .670 | .710 | .712 | .569 | .670 | .673 |
| Semantic Density | .694 | .582 | .628 | .705 | .572 | .635 | .711 | .611 | .617 | .721 | .583 | .624 | .640 |
| RAUQ | **.724** | **.815** | **.713** | .705 | .629 | **.715** | .752 | **.772** | **.718** | **.726** | .597 | **.727** | **.716** |

Table 14: Comparison with supervised methods by PRR↑ for the Llama-3.1 8b model in the in-domain setup across each dataset. The best method is in **bold**, the second best is underlined. Warmer color indicates better results.

| UQ Method | XSum | SamSum | CNN | WMT19 | TruthfulQA | CoQA | SciQ | TriviaQA | MMLU | GSM8k | Mean |
|---|---|---|---|---|---|---|---|---|---|---|---|
| Factoscope | .292 | .064 | -.020 | .120 | .065 | .033 | .313 | .363 | .585 | .121 | .194 |
| SAPLMA | .288 | .382 | .056 | .548 | .277 | -.002 | .399 | .399 | .456 | .358 | .316 |
| MIND | .437 | .361 | .178 | .451 | .411 | .263 | .499 | .517 | **.727** | .570 | .441 |
| Sheeps | .510 | .466 | .380 | .509 | .349 | **.423** | .552 | .594 | .723 | **.604** | .511 |
| LookBackLens | .528 | .441 | .279 | **.613** | .462 | .341 | .542 | .497 | .718 | .525 | .495 |
| SATRMD+MSP | .494 | .495 | .248 | .475 | .448 | .333 | **.581** | .561 | .704 | .528 | .487 |
| TAD | **.550** | **.535** | .444 | .592 | **.463** | .392 | .488 | **.632** | .724 | .557 | **.538** |
| RAUQ | .370 | .464 | **.452** | .509 | .364 | .265 | .506 | .522 | .549 | .323 | .432 |

The results show that in the in-domain experimental setup, supervised methods leveraging attention-based features, such as TAD and LookBackLens, outperform the RAUQ method. Methods that leverage hidden states, such as MIND and Sheeps, achieve performance comparable to RAUQ on average but underperform on the CNN and WMT19 datasets. In contrast, in the out-of-domain experimental setup, RAUQ substantially outperforms on average all supervised methods, which experience a significant performance drop. Our method, however, maintains consistent performance due to its unsupervised nature.

Overall, RAUQ approaches the performance of most supervised methods in in-domain settings, underperforming only those based on attention, while requiring no access to the training dataset. In out-of-domain settings, RAUQ demonstrates a strong advantage, substantially outperforming all supervised approaches.

## D.4 EXPERIMENTS WITH INTERPRETABILITY SCORES

To assess the flexibility and generalization of RAUQ beyond standard LLM architectures with attention layers, we evaluate its performance when original attention weights are replaced with alternative interpretability scores, such as Layer Integrated Gradients (LIG) (Sundararajan et al., 2017).

We conduct an experiment using the LLaMA-3.1-8B model, where we replace the original attention weights with scores derived from Layer Integrated Gradients computed on the output projection layer following the attention module. We manually partition this linear layer in each transformer block into equal segments corresponding to a synthetic division across attention heads, and compute interpretability scores for each segment using Layer Integrated Gradients. This procedure yields matrices analogous to attention weights, preserving the same number of "heads" and layers. We then apply these matrices within the RAUQ method without any modification.

Table 15: Comparison with supervised methods by PRR↑ for the Llama-3.1 8b model in the out-of-domain setup across each dataset. The best method is in **bold**, the second best is underlined. Warmer color indicates better results.

| UQ Method | XSum | SamSum | CNN | WMT19 | TruthfulQA | CoQA | SciQ | TriviaQA | MMLU | GSM8k | Mean |
|---|---|---|---|---|---|---|---|---|---|---|---|
| Factoscope | .105 | .050 | -.065 | .083 | .036 | .014 | .084 | -.017 | .007 | -.040 | .026 |
| SAPLMA | -.035 | .049 | -.009 | -.029 | -.056 | -.020 | -.010 | .224 | -.000 | .152 | .027 |
| MIND | -.077 | .185 | .074 | .158 | .281 | .112 | .166 | .222 | .352 | .316 | .179 |
| Sheeps | .122 | .101 | -.056 | .013 | **.410** | .184 | .365 | .223 | .535 | .310 | .221 |
| LookBackLens | .171 | .197 | .000 | -.018 | .220 | .116 | .285 | .178 | .316 | .189 | .166 |
| SATRMD+MSP | .362 | .098 | **.477** | .364 | .108 | .142 | .190 | .170 | **.572** | .307 | .279 |
| TAD | .269 | .176 | -.101 | .087 | .224 | .143 | .251 | .394 | .432 | **.323** | .220 |
| RAUQ | **.370** | **.464** | .452 | **.509** | .364 | **.265** | **.506** | **.522** | .549 | .323 | **.432** |

The results indicate that RAUQ (LIG) performs comparably to the original RAUQ, with only a negligible performance degradation of 0.4% PRR on average across datasets. These experiments further illustrate that original attention can be effectively substituted with alternative interpretability scores, enabling the application of RAUQ to models without attention mechanisms or with non-standard attention architectures.

Table 16: PRR↑ for Llama-3.1 8b model for RAUQ with original attention weights and with Layer Integrated Gradients (LIG) instead of attention weights. The best method is in **bold**.

| UQ Method | WMT14 | WMT19 | TruthfulQA | CoQA | SciQ | TriviaQA | MMLU | Mean |
|---|---|---|---|---|---|---|---|---|
| RAUQ | **.394** | .509 | **.364** | **.265** | **.506** | **.522** | **.549** | **.444** |
| RAUQ (LIG) | .389 | **.512** | .362 | .264 | .489 | .515 | .547 | .440 |

# E  DETAILED EXPERIMENTAL RESULTS

The detailed experimental results across each considered dataset are presented in Tables 17 to 20 for Llama-3.1 8b, Qwen-2.5 7b, Gemma-2 9b, and Falcon-3 10b models respectively.

Table 17: Detailed PRR↑ for the Llama-3.1 8b model across each dataset. The best method is in **bold**, the second best is underlined. Warmer color indicates better results.

| UQ Method | XSum | SamSum | CNN | WMT14 | WMT19 | MedQUAD | TruthfulQA | CoQA | SciQ | TriviaQA | MMLU | GSM8k | Mean |
|---|---|---|---|---|---|---|---|---|---|---|---|---|---|
| MSP | .313 | .050 | **.525** | .335 | .459 | .091 | .242 | .262 | .459 | .527 | .535 | .310 | .342 |
| Perplexity | **.370** | .456 | .431 | .344 | .416 | **.249** | .377 | .259 | .244 | .506 | .492 | .303 | .370 |
| CCP | .347 | .059 | .514 | .317 | .363 | .038 | .080 | .210 | .351 | .562 | .446 | .306 | .299 |
| Attention Score | .036 | .083 | .258 | .176 | .179 | -.295 | .081 | -.028 | -.142 | .067 | .209 | .209 | .069 |
| Focus | .326 | .281 | .399 | .306 | .416 | .137 | **.380** | .211 | .422 | .507 | .305 | .278 | .331 |
| Simple Focus | .272 | .193 | .454 | .358 | .472 | .074 | .187 | .281 | .486 | .545 | .516 | .302 | .345 |
| DegMat NLI Score entail. | .033 | .147 | .173 | .193 | .285 | .146 | .226 | .316 | .429 | **.583** | .239 | .203 | .248 |
| Ecc. NLI Score entail. | .011 | -.004 | -.031 | .229 | .340 | .102 | .145 | .293 | .380 | .530 | .231 | .235 | .205 |
| EVL NLI Score entail. | .035 | .144 | .164 | .183 | .252 | .137 | .234 | .314 | .371 | .577 | .230 | .188 | .236 |
| Lexical Similarity Rouge-L | .081 | .122 | .190 | .246 | .403 | -.017 | .110 | .277 | .378 | .491 | .242 | .273 | .233 |
| EigenScore | .036 | .130 | .069 | .252 | .318 | -.010 | .079 | .263 | .355 | .462 | .192 | .283 | .202 |
| LUQ | .141 | .221 | .156 | .204 | .224 | .101 | .235 | .303 | .394 | .570 | .249 | .158 | .246 |
| Semantic Entropy | .025 | .105 | .222 | .252 | .379 | .093 | .107 | .232 | .347 | .479 | .157 | **.366** | .230 |
| SAR | .060 | .224 | .227 | .306 | .435 | .107 | .181 | .297 | .439 | .552 | .275 | .320 | .285 |
| Semantic Density | .158 | .154 | .148 | .233 | .295 | .175 | .302 | **.380** | .448 | .571 | .237 | .197 | .275 |
| RAUQ | .370 | **.464** | .452 | **.394** | **.509** | .241 | .364 | .265 | **.506** | .522 | **.549** | .323 | **.413** |

Table 18: Detailed PRR↑ for the Qwen-2.5 7b model across each dataset. The best method is in **bold**, the second best is underlined. Warmer color indicates better results.

| UQ Method | XSum | SamSum | CNN | WMT14 | WMT19 | MedQUAD | TruthfulQA | CoQA | SciQ | TriviaQA | MMLU | GSM8k | Mean |
|---|---|---|---|---|---|---|---|---|---|---|---|---|---|
| MSP | .088 | -.003 | .368 | .286 | .451 | .030 | -.101 | .291 | .551 | .610 | **.654** | .268 | .291 |
| Perplexity | .242 | **.289** | .229 | **.346** | .466 | **.131** | .156 | .270 | .385 | .601 | .400 | .456 | .331 |
| CCP | .243 | .021 | .294 | .266 | .388 | .015 | -.089 | .215 | .468 | .596 | .412 | .281 | .259 |
| Attention Score | .037 | .103 | .250 | .136 | .149 | .022 | -.023 | .007 | -.105 | .078 | .157 | .131 | .078 |
| Focus | .214 | .149 | .196 | .308 | .452 | .123 | .137 | .249 | .462 | .568 | .037 | .273 | .264 |
| Simple Focus | .117 | .086 | .205 | .302 | .496 | .021 | .037 | .321 | .536 | .620 | .550 | .310 | .300 |
| DegMat NLI Score entail. | .141 | .178 | .145 | .217 | .332 | .122 | .293 | .329 | .540 | .574 | .235 | .402 | .292 |
| Ecc. NLI Score entail. | -.058 | .044 | .021 | .243 | .368 | .107 | .151 | .294 | .535 | .543 | .237 | .386 | .239 |
| EVL NLI Score entail. | .141 | .183 | .138 | .196 | .294 | .122 | .294 | .329 | .519 | .571 | .236 | .372 | .283 |
| Lexical Similarity Rouge-L | .119 | .161 | .112 | .284 | .370 | .075 | .141 | .297 | .507 | .531 | .274 | .511 | .282 |
| EigenScore | .079 | .034 | .071 | .231 | .374 | .018 | -.003 | .281 | .510 | .500 | .243 | .537 | .240 |
| LUQ | .224 | .260 | .104 | .161 | .265 | .096 | **.340** | .337 | .449 | .580 | .321 | .331 | .289 |
| Semantic Entropy | .021 | .109 | .146 | .268 | .366 | .073 | .058 | .265 | .491 | .536 | .165 | .380 | .240 |
| SAR | .128 | .186 | .145 | .340 | .445 | .088 | .196 | .318 | .526 | .585 | .288 | .459 | .309 |
| Semantic Density | .084 | .156 | .092 | .225 | .358 | .095 | .285 | **.386** | .514 | .603 | .203 | .381 | .282 |
| RAUQ | .180 | .206 | .254 | .344 | **.533** | .123 | -.020 | .252 | .499 | .608 | .584 | .458 | **.335** |

Table 19: Detailed PRR↑ for the Gemma-2 9b model across each dataset. The best method is in **bold**, the second best is underlined. Warmer color indicates better results.

| UQ Method | XSum | SamSum | CNN | WMT14 | WMT19 | MedQUAD | TruthfulQA | CoQA | SciQ | TriviaQA | MMLU | GSM8k | Mean |
|---|---|---|---|---|---|---|---|---|---|---|---|---|---|
| MSP | .333 | .095 | **.574** | .279 | .484 | .004 | .152 | .310 | .501 | .649 | .599 | .310 | .357 |
| Perplexity | .329 | .308 | .488 | .362 | .449 | .397 | .240 | .314 | .234 | **.660** | .578 | .256 | .385 |
| CCP | **.407** | .061 | .566 | .270 | .369 | .028 | .092 | .277 | .385 | .633 | .550 | .339 | .332 |
| Attention Score | -.043 | .061 | .291 | .131 | .161 | -.150 | .083 | -.016 | -.112 | .075 | .300 | .268 | .087 |
| Focus | .276 | .308 | .436 | .305 | .465 | **.514** | .230 | .289 | .434 | .619 | .563 | .265 | .392 |
| Simple Focus | .258 | .169 | .538 | .324 | .521 | .170 | .238 | .335 | .523 | .656 | .570 | .280 | .382 |
| DegMat NLI Score entail. | .061 | .232 | .120 | .206 | .312 | .167 | .141 | .312 | .422 | .619 | .401 | .293 | .274 |
| Ecc. NLI Score entail. | -.000 | .072 | -.012 | .237 | .343 | .037 | .132 | .299 | .419 | .569 | .399 | .228 | .227 |
| EVL NLI Score entail. | .062 | .231 | .105 | .202 | .302 | .176 | .159 | .304 | .389 | .615 | .398 | .284 | .269 |
| Lexical Similarity Rouge-L | .059 | .168 | .257 | .279 | .404 | -.035 | .113 | .319 | .395 | .585 | .418 | .346 | .276 |
| EigenScore | .016 | .082 | .221 | .204 | .249 | -.024 | .132 | .270 | .359 | .519 | .371 | .241 | .220 |
| LUQ | .199 | .247 | .172 | .242 | .276 | .222 | .250 | .301 | .342 | .618 | .440 | .237 | .295 |
| Semantic Entropy | .013 | .101 | .263 | .273 | .401 | .083 | .026 | .265 | .355 | .551 | .427 | .328 | .257 |
| SAR | .084 | .289 | .331 | .373 | .455 | .203 | .166 | .323 | .362 | .626 | .493 | **.355** | .338 |
| Semantic Density | .163 | .149 | .188 | .196 | .313 | .272 | **.357** | **.401** | .463 | .654 | .295 | .183 | .303 |
| RAUQ | .329 | **.340** | .508 | **.391** | **.554** | .331 | .257 | .331 | .481 | .633 | **.628** | .283 | **.422** |

Table 20: Detailed PRR↑ for the Falcon-3 10b model across each dataset. The best method is in **bold**, the second best is underlined. Warmer color indicates better results.

| UQ Method | XSum | SamSum | CNN | WMT14 | WMT19 | MedQUAD | TruthfulQA | CoQA | SciQ | TriviaQA | MMLU | GSM8k | Mean |
|---|---|---|---|---|---|---|---|---|---|---|---|---|---|
| MSP | .178 | .053 | **.301** | .269 | .396 | -.004 | -.001 | .300 | .459 | .674 | .621 | .364 | .301 |
| Perplexity | .141 | .152 | .248 | .355 | .524 | **.266** | .209 | .276 | .158 | .660 | .617 | .307 | .326 |
| CCP | .128 | .043 | .213 | .249 | .325 | -.041 | -.002 | .259 | .349 | .653 | .533 | .339 | .254 |
| Attention Score | **.272** | .077 | .227 | .113 | .064 | -.037 | -.024 | -.034 | -.073 | .109 | .226 | .210 | .094 |
| Focus | .159 | .069 | .187 | .262 | .463 | .123 | .208 | .218 | .304 | .656 | .486 | .195 | .278 |
| Simple Focus | .089 | .046 | .150 | .313 | .457 | .005 | .160 | .325 | .388 | **.680** | .603 | .294 | .292 |
| DegMat NLI Score entail. | .107 | .152 | .136 | .140 | .304 | .115 | .203 | .326 | .391 | .617 | .418 | **.391** | .275 |
| Ecc. NLI Score entail. | -.028 | .104 | .037 | .203 | .360 | .097 | .066 | .298 | .432 | .593 | .437 | .368 | .247 |
| EVL NLI Score entail. | .103 | **.157** | .145 | .131 | .281 | .111 | .204 | .319 | .436 | .618 | .403 | .366 | .273 |
| Lexical Similarity Rouge-L | .096 | .090 | .065 | .211 | .339 | .035 | .087 | .306 | .238 | .595 | .454 | .281 | .233 |
| EigenScore | .064 | .010 | .079 | .177 | .294 | -.067 | .104 | .283 | .336 | .542 | .357 | .173 | .196 |
| LUQ | .134 | .134 | .095 | .126 | .265 | .127 | .237 | .307 | .270 | .622 | .423 | .358 | .258 |
| Semantic Entropy | .143 | .102 | .153 | .222 | .361 | .026 | .102 | .301 | .379 | .587 | .462 | .381 | .268 |
| SAR | .084 | .119 | .079 | .256 | .419 | .070 | .180 | .308 | .253 | .650 | .514 | .364 | .275 |
| Semantic Density | .129 | .155 | .139 | .208 | .352 | .075 | **.272** | **.350** | **.524** | .620 | .352 | .314 | .291 |
| RAUQ | .151 | .156 | .235 | **.376** | **.553** | .224 | .110 | .292 | .474 | .674 | **.626** | .344 | **.351** |

# F    ADDITIONAL EXAMPLES

We provide more examples of attention maps, similar to the Figure 1, in Figures 6 to 9. These examples show that the similar patterns exist for several text instances.

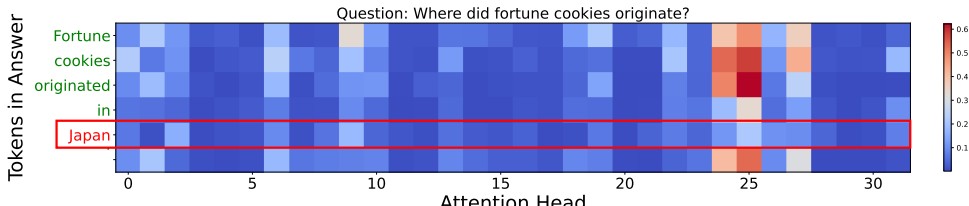

Figure 6: Attention weights in the 30th layer of Llama-3.1 8B from each generated token to its preceding token, given the prompt *Where did fortune cookies originate?*. The $y$ axis specifies the generated tokens, and the $x$ axis specifies the attention heads. Warmer colors indicate higher attention values. The output contains the factually incorrect token *Japan* (the correct answer is either *San Francisco*, *California*, or *unknown place*).

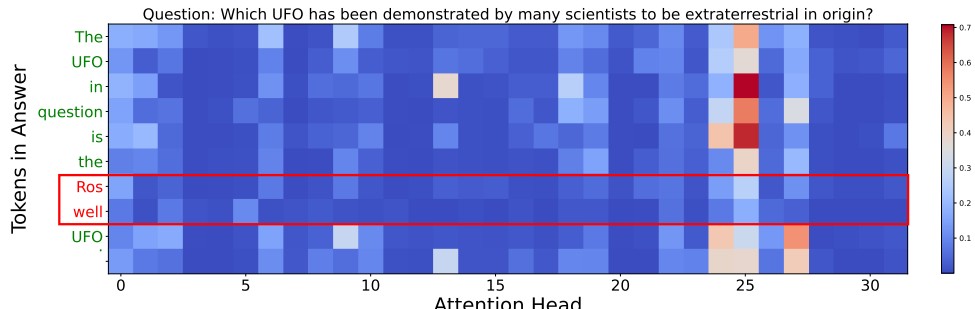

Figure 7: Attention weights in the 30th layer of Llama-3.1 8B from each generated token to its preceding token, given the prompt *Which UFO has been demonstrated by many scientists to be extraterrestrial in origin?*. The $y$ axis specifies the generated tokens, and the $x$ axis specifies the attention heads. Warmer colors indicate higher attention values. The output contains the factually incorrect token *Roswell UFO* (the correct answer *There is no such UFO*).

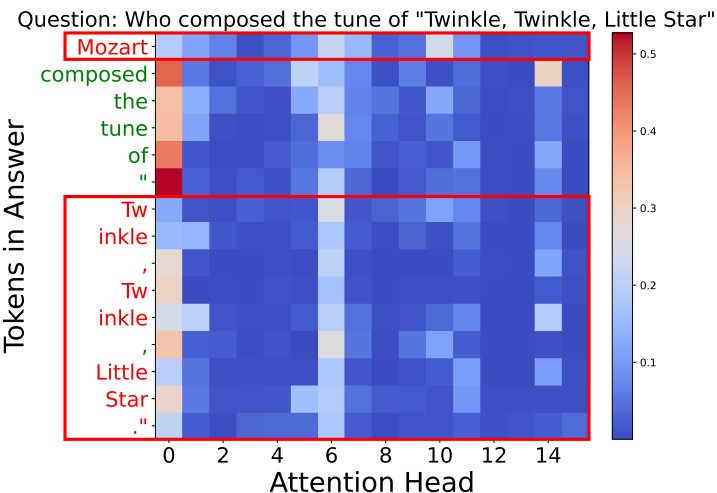

Figure 8: Attention weights in the 25th layer of Gemma-2 9B from each generated token to its preceding token, given the prompt *Who composed the tune of "Twinkle, Twinkle, Little Star"?*. The $y$ axis specifies the generated tokens, and the $x$ axis specifies the attention heads. Warmer colors indicate higher attention values. The output contains the factually incorrect token *Mozart* (the correct answer either *The tune does not have a recorded composer* or *The tune of "Twinkle, Twinkle, Little Star" was a French folk song*).

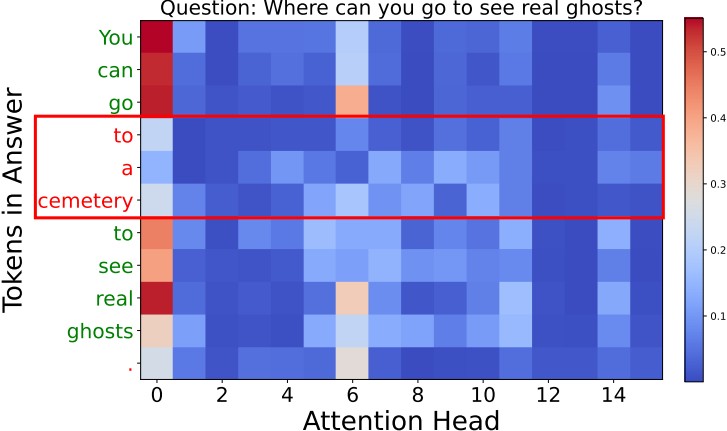

Figure 9: Attention weights in the 25th layer of Gemma-2 9B from each generated token to its preceding token, given the prompt *Where can you go to see real ghosts?*. The $y$ axis specifies the generated tokens, and the $x$ axis specifies the attention heads. Warmer colors indicate higher attention values. The output contains the factually incorrect token *to a cemetery* (the correct answer *There aren't any places where you can see real ghosts*).

# G   ERROR ANALYSIS

To further investigate which generations are chosen by RAUQ, we conducted an error analysis on a small subset of the TruthfulQA dataset. To do so, we chose the top-20 samples with the highest and lowest RAUQ scores and carefully attributed the corresponding generations as truthful or erroneous. For errors, we also analyzed each error as a factual or reasoning error. The results are presented in Table 21.

Table 21: Error analysis for detected by RAUQ generations for Llama-3.1 8b on TruthfulQA dataset.

|  | Erroneous generations (reasoning / factual) | Truthful generations |
|---|---|---|
| Samples with highest RAUQ scores | 95% (35% / 60%) | 5% |
| Samples with lowest RAUQ scores | 50% (15% / 35%) | 50% |

## H    LIMITATIONS

Our approach is unsupervised and involves only a single hyperparameter. While we demonstrate that a predefined value yields robust performance across various tasks, fine-tuning this parameter for specific datasets could lead to further improvements, which would require a validation set.

In this work, we focus on white-box UQ methods – techniques that assume full access to the internal states of an LLM. Although such methods cannot be directly applied to black-box models (e.g. LLMs exposed only through API), our work demonstrates that white-box access enables substantially performance improvements, while remaining computationally efficient. Consequently, our approach paves the way for integrating robust UQ mechanisms directly into existing LLM-as-a-service systems, which is highly useful for real-world applications.

Nevertheless, one possible direction for adapting our technique to a black-box setting is to employ an auxiliary white-box proxy LLM from which attention signals and logits can be extracted. Such a proxy model may be effective because it can detect ambiguous or underspecified queries, thereby capturing uncertainty patterns that partially mirror those of the black-box target model.

