# OpenReview forum: "Efficient Hallucination Detection for LLMs Using Uncertainty-Aware Attention Heads"
_ICLR.cc/2026/Conference — Submitted to ICLR 2026_

### Official Review · Reviewer_KfLE · 2025-10-28

**Soundness:** 3
**Presentation:** 3
**Contribution:** 2
**Rating:** 6
**Confidence:** 3

**Summary:**

This paper proposes RAUQ (Recurrent Attention-based Uncertainty Quantification), an unsupervised white-box method for hallucination detection. The key insight is that certain "uncertainty-aware" attention heads reduce focus on preceding tokens when models generate incorrect information. RAUQ identifies these heads per layer, computes token-level confidence scores recurrently combining attention weights and token probabilities, and aggregates to sequence-level uncertainty using maximum across layers. Evaluated on 4 LLMs across 12 diverse tasks (QA, summarization, translation), RAUQ achieves state-of-the-art performance with <1% computational overhead.

**Strengths:**

The empirical results are impressive and consistent. RAUQ substantially outperforms 15 baselines across diverse tasks and models, with particularly strong gains on translation tasks (mean PRR 0.384 vs. 0.326 for the next-best method). The mechanistic insights are compelling: Figures 1-4 clearly show how certain attention heads reduce focus on preceding tokens precisely at hallucination points, providing genuine intuition for why the method works.

The practical efficiency is exceptional—less than 1% computational overhead compared to 400-800% for sampling-based competitors. A single forward pass requirement makes deployment genuinely feasible. The experimental evaluation is thorough and comprehensive, encompassing 4 LLM families across 12 diverse datasets with detailed ablation studies examining aggregation functions, hyperparameters, and layer selection. Findings generalize across tasks without task-specific tuning, and middle layers consistently emerge as most informative. The method is reproducible with promised code release and detailed experimental specifications.

The work demonstrates robustness: performance holds across different model families and diverse generation tasks without modification. The analysis shows that unsupervised uncertainty emerges naturally in model internals, providing theoretical encouragement that these patterns reflect genuine uncertainty rather than artifacts.

**Weaknesses:**

The theoretical foundation is shallow. The paper shows that specific attention heads drop focus at hallucination tokens but doesn't deeply explain the underlying mechanism. Why does maximum aggregation across layers emerge as optimal? Why does combining head selection, recurrence, and aggregation work together synergistically? Limited ablations on component interactions make it hard to understand which pieces are essential. The connection between attention patterns and hallucinations relies on empirical observation rather than principled explanation.

Several methodological choices appear ad-hoc. Head selection based on average attention to previous tokens lacks justification—alternative metrics aren't considered. The fixed decision center (top-right) is used across all models and tasks despite the paper acknowledging optimal centers vary by context. The hyperparameter α=0.2 is selected via validation on Llama-2 but robustness across all evaluated models isn't verified. Layer selection (first to second third) also seems arbitrary, especially since Table 8 suggests individual layers sometimes outperform the aggregated version.

The evaluation uses inconsistent quality metrics across tasks (accuracy for QA, AlignScore for summarization, COMET for machine translation), making unified interpretation difficult. There's no evaluation on instruction-following, dialog, or diverse generation types. Critically, selected heads differ substantially across datasets (Tables 9-10), which is concerning for a method claiming to be truly unsupervised. The paper lacks analysis of which error types (factual, reasoning, formatting) are detected or why answer length significantly affects performance across tasks.

Generalization is questionable. Only open-source models are evaluated; whether closed-source models (GPT-4, Claude) exhibit the same attention patterns remains unknown. There's no evaluation on models trained with different objectives (DPO, preference learning) or on a wider size range (no models <3B or >70B). Tables 12-13 show supervised methods substantially outperform RAUQ in-domain, suggesting a real opportunity cost to the unsupervised approach.

White-box access is required, limiting applicability since most users access models through APIs. The method's reliance on specific attention patterns may not generalize to future architectures with different attention mechanisms. The core assumption that hallucinations cause detectable attention drops hasn't been validated on adversarial inputs where models hallucinate confidently.

**Questions:**

1. Why specifically does attention to previous token drop at hallucinations? Have you verified this pattern holds for other attention patterns (e.g., attention to question)?

2. Can you provide theoretical justification for the recurrent confidence formula (Equation 2)? Why is this the right way to propagate uncertainty?

3. How does performance vary if you adapt α per-model using validation data?

4. Why do selected heads differ substantially across datasets if the method is fully unsupervised?

5. How does the method perform on adversarial inputs where models hallucinate confidently?

6. Can you test on models with non-standard attention mechanisms (e.g., sparse attention, linear attention)?

---

> ### Author Response · Authors · 2025-11-23
>
> Thank you very much for your time in reviewing our paper and your very valuable comments.
>
> **Q1. Regarding theoretical motivation. The paper shows that specific attention heads drop focus at hallucination tokens but doesn't deeply explain the underlying mechanism. … The connection between attention patterns and hallucinations relies on empirical observation rather than principled explanation**
>
> A1. The theoretical motivation of RAUQ builds on prior attention-based UQ methods, such as AttentionScore (LLM-Check) (Sriramanan et al., NeurIPS 2024), StrongerFocus (Zhang et al., 2023), and TAD (Vazhentsev et al, 2025).
>
> Sriramanan et al. (2024) illustrate that attention weights contain patterns indicative of hallucinations through eigen-analysis of attention kernels. They justify using only the attention weights to the previous token, as these correspond to the eigenvalues of the lower triangular attention matrix, and their sum exactly equals its log determinant. In our work, we reveal a similar pattern through a mechanistic analysis of attention weights, examining the correlation between hallucinations and attention weight distributions.
>
> Zhang et al. (2023) and Vazhentsev et al. (2025) illustrate that spotting LLM hallucinations require recurrent uncertainty propagation from previous generation steps as probability distribution modeled by the LLM is a conditional distribution. We directly leverage this theoretical finding in the recurrent aggregation formula (2).
>
> Our main heuristic: attention head selection is based on our observation that the majority of heads are not indicative of hallucinations (Figure 3a). Our approach simply selects the most contrastive head that has the best potential for discriminating between hallucinations and non-hallucinations. Our findings are well supported by prior mechanistic interpretability studies of attention heads, which have shown that different heads serve distinct functions (Elhelo et al. 2025). Thus, our empirical finding is consistent with previous research.
>
> Thus, RAUQ is well-motivated as it builds on theoretical findings in previous work. Moreover, our insights helped to improve the previous attention-based method AttentionScore. Section 5.3, lines 493-505, shows that integrating RAUQ components into AttentionScore (Sriramanan et al., 2024) significantly boosts its performance.
>
> To address your concern, we have added a subsection in Section 4 that provides a broader discussion of the theoretical grounding of the method’s components.
>
> **Q2. Why does combining head selection, recurrence, and aggregation work together synergistically?**
>
> A2. The components of RAUQ can be used not only in combination, but individually as well.
> In the experiment, presented in Section 5.3 (lines 493-505), we show how different components independently contribute to the performance of another attention-based method -- AttentionScore (Sriramanan et al., NeurIPS 2024). The results show that RAUQ components individually boost the performance of RAUQ and AttentionScore.
>
> **Q3. Why does maximum aggregation across layers emerge as optimal?**
>
> A3. Maximum provides an upper bound on uncertainty. By taking the maximum across attention heads within each layer and then across layers, our method simply selects the most contrastive attention head that has the best potential for discriminating between hallucinations and non-hallucinations.
>
> We also validate this approach through ablation study presented in Section 5.3 and Table 6. It shows that alternative aggregation strategies result in worse performance.

---

> ### Author Response · Authors · 2025-11-23
>
> **Q4. Limited ablations on component interactions make it hard to understand which pieces are essential.**
>
> A4. We conduct a thorough ablation studies of components, and individual design decisions, such as aggregations. In Table 7, we conduct experiments with several modifications of the recurrent formulas used to compute token-level confidence scores. We consider: (1) removing attention weights, (2) removing recurrence, (3) replacing the previous token’s confidence score with its probability, (4) multiplying probabilities with attention weights, and (5) the recurrent formula proposed in RAUQ. This ablation essentially represents the interactions among different components of our method.
>
> We observe that multiplying probabilities with attention weights without recurrence (4) and removing attention weights (1) both lead to significant performance degradation, indicating that recurrence and attention mechanisms are the most influential components of the method. We also find that removing recurrent confidence (2) and replacing recurrent confidence with probabilities (3) also result in worse performance, demonstrating the importance of uncertainty propagation.
>
> Additionally, we study how individual components can contribute to the performance of another attention-based method -- AttentionScore (Sriramanan et al., 2024). Section 5.3, lines 493-505, shows the impact of integrating individual RAUQ components into AttentionScore. All components give substantial boost in performance, especially the attention head selection.
>
> We also conduct various studies on aggregating the token-level confidence scores and layer-wise uncertainty scores, as shown in Tables 5 and 6 in Appendix C.1.
>
> Overall, we believe that the paper clearly illustrates the interactions of the components of the proposed method. We would be grateful for any particular suggestions to extend the ablation studies further.
>
> **Q5. Head selection based on average attention to previous tokens lacks justification—alternative metrics aren't considered.**
>
> A5. The aggregation methods used in RAUQ represent simple standard practices. Token-level scores are aggregated in a manner similar to perplexity, while layer-wise scores are aggregated using the maximum to provide an upper bound on the estimated uncertainty. We ablate these design choices through ablation studies presented in Section 5.3 and Tables 5-7, which show that alternative aggregation strategies result in worse performance compared to the ones we use.
>
> **Q6. The fixed decision center (top-right) is used across all models and tasks despite the paper acknowledging optimal centers vary by context. The hyperparameter α=0.2 is selected via validation on Llama-2 but robustness across all evaluated models isn't verified.**
>
> A6. We respectfully disagree. We use the same hyperparameter $\alpha=0.2$ across all tasks and LLMs exactly to demonstrate robustness of RAUQ and that it does not need careful tuning of hyperparameters for each task and LLM. After we selected the hyperparameter for one task for Llama-2, we used it in all experiments. All Table 1 results use fixed $\alpha=0.2$ and consistently outperform the baselines across 12 datasets, 3 task types and 4 LLMs.
>
> Additionally, we conduct an ablation study, where we carefully tune $\alpha$ and obtain optimal value for each specific task and LLM. As illustrated in Figure 5, the optimal $\alpha$ consistently falls between 0.1 and 0.3 for all cases except MMLU, which is close to our choice $\alpha=0.2$, and our choice usually yields near optimal performance.
>
> Therefore, we show robustness across all evaluated models: RAUQ  can be used in a plug-and-play fashion with a fixed $\alpha$, eliminating the need for task- or model-specific tuning.
>
> **Q7. Layer selection (first to second third) also seems arbitrary, especially since Table 8 suggests individual layers sometimes outperform the aggregated version.**
>
> A7. We use intermediate layers because prior work (Azaria et al., 2023; Vazhentsev et al., 2025) shows they are most informative for hallucination detection. Importantly, we use the same fixed range (first to second third) of layers for all LLMs without any tuning.
>
> Our ablation study on layer choice in Table 8 in Appendix C.2 shows that our approach is a highly robust unsupervised strategy. Selecting optimal layers offers only marginal improvements, yet it requires supervision: tuning the layer for each task on a validation set.

---

> ### Author Response · Authors · 2025-11-23
>
> **Q8. The evaluation uses inconsistent quality metrics across tasks (accuracy for QA, AlignScore for summarization, COMET for machine translation), making unified interpretation difficult.**
>
> A8. We follow the standard practice for benchmarking UQ methods (Vashurin et al. 2025, Santilli et al. 2025, Vazhentsev et al. 2025). As a generation quality metric, we use the metric most appropriate for each task, since a single quality metric usually is not applicable across tasks such as MT and QA.
>
> Nevertheless, for assessing the quality of UQ methods, we use the single metric PRR (with different generation quality metrics), which is comparable across tasks as it reflects how effectively the UQ methods identify factual errors produced by the models.
>
> **Q9. There's no evaluation on instruction-following, dialog, or diverse generation types.**
>
> A9. We follow standard practices for benchmarking UQ methods (Vashurin et al. 2025, Santilli et al. 2025, Vazhentsev et al. 2025). Our experiments encompass 12 datasets across 3 task types, covering a wide range of generation types.
>
> **Q10. Critically, selected heads differ substantially across datasets (Tables 9-10), which is concerning for a method claiming to be truly unsupervised.**
>
> A10. RAUQ does not need validation data to select attention heads, as a built-in procedure does that in an unsupervised manner. Tables 9 and 10 in Appendix C.2 present the ablation study, analyzing the selected attention heads for WMT14 De-En and CoQA. Moreover, this analysis proves the stability of the selected heads. When comparing the chosen heads across the two datasets, we observe a substantial degree of overlap. Notably, in layers 10, 12, 13, 15, 16, and 20, the selected heads are perfectly aligned, indicating strong consistency across tasks. Thus, although some variation exists, the same heads generally provide the most informative signals for the RAUQ method, highlighting both intra-task and cross-task stability.
>
> Our analysis shows that the method does not require any supervision (validation data) to be effective and can be used in plug-and-play fashion.
>
> **Q11. The paper lacks analysis of which error types (factual, reasoning, formatting) are detected or why answer length significantly affects performance across tasks**
>
> A11. We analyzed samples with the highest and the lowest RAUQ scores for LLaMA-3.1 8B on TruthfulQA dataset. RAUQ effectively detects erroneous generations, with most of the detected errors attributed as factual and reasoning errors. As we use relatively new models, there are almost no formatting errors in generations. However, not all of the generations with low uncertainty are truthful. Most of the erroneous generations with low uncertainty correspond to the common misbeliefs. Here is an example:
> Question: _ If you're visiting Germany and Sweden, you'll need to change dollars into what? _
> Generated answer: _ Euros _
> Ground truth answer: _ Euro and krona _
> This is an example with common misbelief, in which the model shows high confidence during generation.
>
> We have introduced the analysis of errors detected by UQ in the new version of the paper in Section 5.3.
>
> The answer length might affect the performance across tasks, as there might be natural bias in quality for shorter or longer lengths (Vashurin et al. 2025). Vashurin et al. 2025 suggest a general approach to dealing with this bias.

---

> ### Author Response · Authors · 2025-11-23
>
> **Q12. Generalization is questionable. Only open-source models are evaluated; whether closed-source models (GPT-4, Claude) exhibit the same attention patterns remains unknown. There's no evaluation on models trained with different objectives (DPO, preference learning) or on a wider size range (no models <3B or >70B).**
>
> A12. Since GPT-4 and Claude are closed-source, naturally, it is impossible to conduct investigation of white-box methods on these LLMs, as such a group of techniques require full access to LLM parameters.
>
> Our experiments were conducted using widely adopted middle sized LLMs. However, because architectures and pre-training procedures are largely consistent across model sizes, RAUQ generalizes effectively to both larger and smaller LLMs.
> To further demonstrate the robustness of RAUQ, we conduct additional experiments using the SmolLM2-360M, Llama-3.2-1B, and LLaMA-3.1-70B models. For models ≤1B parameters, we exclude MMLU, GSM8K, and MedQUAD due to their near-zero performance on these tasks. We compare RAUQ against the most prominent baselines identified in Table 1 and provide here a table in the same format. The results show that RAUQ is the best method for QA and MT on ≤1B LLMs, and for Summ and MT on 70B LLM. Overall, RAUQ surpasses the second-best method by an average of 4% across all tasks and models. These results highlight the strong generalization ability of RAUQ across a wide range of model sizes.
>
> We have included this experiment and the corresponding table in the new version of the paper in Table 12 in Appendix D.1.
>
> | Methods | SmolLM2-360M -- QA | SmolLM2-360M -- Summ | SmolLM2-360M -- MT | Llama-3.2-1B -- QA | Llama-3.2-1B -- Summ | Llama-3.2-1B -- MT | Llama-3.1-70B -- QA | Llama-3.1-70B -- Summ | Llama-3.1-70B -- MT | Mean |
> |:--------|-------------------:|--------------------:|--------------------:|-------------------:|--------------------:|--------------------:|---------------------:|----------------------:|----------------------:|------:|
> | MSP               |                                   0.36 |                                    **0.45** |                                    0.33 |                             0.32 |                              **0.51** |                              0.35 |                                 0.36 |                                  0.13 |                                  0.45 |           0.36 |
> | Perplexity        |                                   0.37 |                                    0.33 |                                    0.49 |                             0.31 |                              0.39 |                              0.43 |                                 0.32 |                                  0.25 |                                  0.34 |           0.36 |
> | Attention Score   |                                   0.07 |                                    0    |                                    0.12 |                             0.05 |                              0.03 |                              0.1  |                                 0.05 |                                  0.05 |                                  0.21 |           0.08 |
> | SAR               |                                   0.34 |                                    0.09 |                                    0.35 |                             0.3  |                              0.09 |                              0.41 |                                 0.38 |                                  0.11 |                                  0.37 |           0.27 |
> | Semantic Density  |                                   0.36 |                                    0.21 |                                    0.26 |                             0.35 |                              0.22 |                              0.28 |                                 **0.39** |                                  0.1  |                                  0.24 |           0.27 |
> | RAUQ              |                                   **0.43** |                                    0.36 |                                   **0.49** |                             **0.36** |                              0.42 |                              **0.49** |                                 0.36 |                                  **0.25** |                                  **0.46** |           **0.4**  |

---

> ### Author Response · Authors · 2025-11-23
>
> **Q13. Tables 12-13 show supervised methods substantially outperform RAUQ in-domain, suggesting a real opportunity cost to the unsupervised approach.**
>
> A13. Supervised UQ methods rely on additional annotated data and therefore tend to overfit to the training domain. While they may achieve strong in-domain performance, they often struggle to generalize beyond it (Vazhentsev et al., 2025), which poses a major challenge for their use in real-world settings.
>
> In practice, OOD performance is far more critical, as real-world applications inevitably involve domain shifts. RAUQ, being an unsupervised approach, avoids domain-specific overfitting and consistently outperforms supervised methods on average in OOD scenarios, underscoring its practical utility in real scenarios.
>
> **Q14. White-box access is required, limiting applicability since most users access models through APIs.**
>
> A14. Our work illustrates that white-box access enables substantial performance improvements while keeping the method highly computationally efficient. Consequently, our approach paves the way for integrating robust UQ mechanisms directly into existing LLM-as-a-service systems, which is highly useful for real-world applications.
>
> Nevertheless, one possible way of adapting our technique to a black-box setting (LLMs through API) is to employ an auxiliary white-box proxy LLM, from which attention signals and logits can be extracted. Applying such a proxy model can be effective because it allows detecting ambiguous or underspecified queries, thereby capturing uncertainty patterns that partially mirror those of the black-box target model.
>
> We have expanded the discussion on the black-box applicability of RAUQ in the Limitations section of the new version of the paper.
>
> **Q15. The method's reliance on specific attention patterns may not generalize to future architectures with different attention mechanisms.**
>
> We could adapt RAUQ to architectures with different attention mechanisms by substituting the attention weights with alternative interpretability scores, such as Layer Integrated Gradients or other techniques (Ali et al. 2025). The major principles will remain the same. (see also A23 for additional ablation study).
>
> **Q16. The core assumption that hallucinations cause detectable attention drops hasn't been validated on adversarial inputs where models hallucinate confidently.**
>
> Our experiments span 12 datasets across 3 task types, demonstrating robustness over a wide range of tasks and inputs. Moreover, to validate performance under adversarial inputs, we also conduct experiments with the TruthfulQA dataset. TruthfulQA is primarily driven by model misconceptions, which may lead to confident hallucinations. The results in Tables 17-20 in Appendix E show the competitive performance of the RAUQ method for such a task.
>
> **Q17. Why specifically does attention to previous token drop at hallucinations?**
>
> A17. We hypothesize that this pattern may arise from the tendency of LLMs to distribute attention more broadly across preceding tokens when the model is uncertain. Sriramanan et al. (2024) show that attention weights exhibit patterns indicative of hallucinations through an eigen-analysis of attention kernels. They justify focusing solely on attention weights to the immediately preceding token, as these correspond to the eigenvalues of the lower-triangular attention matrix, and their sum equals its log determinant. In our work, we identify a similar pattern through a mechanistic analysis of attention weights, examining the relationship between hallucinations and attention weight distributions (Figure 3).
>
> **Q18. Have you verified this pattern holds for other attention patterns (e.g., attention to question)?**
>
> A18. We find that attention to later tokens does not correlate with hallucinations, as the model tends to spread attention more broadly across previous tokens when uncertain (Figure 4). Consequently, attention to the question itself should not be expected to correlate with hallucination behavior.
>
> **Q19. Can you provide theoretical justification for the recurrent confidence formula (Equation 2)? Why is this the right way to propagate uncertainty?**
>
> A19. The theoretical justification for introducing recurrence into the computation of uncertainty scores builds on Zhang et al. (2023) and Vazhentsev et al. (2025). These works show that effective hallucination detection requires recurrent propagation of uncertainty across generation steps, because the probability distribution modeled by an autoregressive LLM is inherently conditional. We explicitly leverage this insight in the recurrent aggregation in Eq. (2), where attention weights act as natural signals for modeling conditional dependencies between successive generation steps.

---

> ### Author Response · Authors · 2025-11-23
>
> **Q20. How does performance vary if you adapt α per-model using validation data?**
>
> A20. As shown in Figure 5, performance can vary slightly when $\alpha$ is optimized per model and task using validation data. While this tuning may provide small gains, it requires additional supervision, whereas our method is designed to be fully unsupervised. All reported RAUQ results use a fixed $\alpha = 0.2$. RAUQ consistently delivers robust improvements over the baselines across 12 datasets, 3 task types, and 4 different models.
>
> **Q21. Why do selected heads differ substantially across datasets if the method is fully unsupervised?**
>
> A21. The most informative heads for RAUQ are largely stable. Our ablation study on head selection in Tables 9 and 10 in Appendix C.2 reveals notable consistency in the selected heads across datasets. Moreover, selected heads across different tasks are also consistent. For example, in layers 10, 12, 13, 15, 16, and 20, many similar heads are selected for CoQA and WMT14 De-En tasks.
>
> Note that while RAUQ selects attention heads independently for each input sequence, the procedure remains fully unsupervised: it simply applies the argmax rule in Eq. 1 without relying on any validation data. This design keeps the method plug-and-play and applicable to any new task or model without supervision.
>
> **Q22. How does the method perform on adversarial inputs where models hallucinate confidently?**
>
> A22. Experiments on the TruthfulQA dataset (Tables 17–20 in Appendix E) illustrate the performance of RAUQ under adversarial input settings. In this task, LLM errors are primarily driven by model misconceptions, often arising from ambiguity in the training data that leads to confident hallucinations. The results show that, in majority cases, RAUQ performs competitively with sampling-based methods, which require substantially higher computational cost.
>
> **Q23. Can you test on models with non-standard attention mechanisms (e.g., sparse attention, linear attention)?**
>
> A23. We could adapt RAUQ to these models by substituting the attention weights with alternative interpretability scores, such as Layer Integrated Gradients or other techniques (Ali et al. 2025). The major principles will remain the same.
>
> To demonstrate the effectiveness of RAUQ under this setting, we conduct an experiment using the LLaMA-3.1-8B model, where we replace the original attention weights with scores derived from Layer Integrated Gradients computed on the output projection layer following the attention module. We manually partition this linear layer in each transformer block into equal segments corresponding to a synthetic division across attention heads, and compute interpretability scores for each segment using Layer Integrated Gradients. This procedure yields matrices analogous to attention weights, preserving the same number of “heads’’ and layers. We then apply these matrices within the RAUQ method without any modification.
>
> The results indicate that RAUQ (LiG) performs comparably to the original RAUQ, with only a negligible performance degradation of 0.4% PRR on average across datasets. These experiments further illustrate that original attention can be effectively substituted with alternative interpretability scores, enabling the application of RAUQ to models without attention mechanisms or with non-standard attention architectures.
>
> We have included this experiment and the corresponding table in the new version of the paper in Section 5.3 and Table 16 in Appendix D.4.
>
> | Method        | WMT14 | WMT19 | TruthfulQA | CoQA | SciQ | TriviaQA | MMLU | Mean |
> |:--------------|------:|------:|-----------:|-----:|-----:|---------:|-----:|-----:|
> | RAUQ          | **0.394** | 0.509 | **0.364** | **0.265** | **0.506** | **0.522** | **0.549** | **0.444** |
> | RAUQ (LiG)    | 0.389 | **0.512** | 0.362 | 0.264 | 0.489 | 0.515 | 0.547 | 0.440 |
>
>
> **References:**
>
> [1] Ali et al. The Hidden Attention of Mamba Models. ACL 2025. \
> [2] Azaria & Mitchell. The Internal State of an LLM Knows When It’s Lying. EMNLP Findings 2023. \
> [3] Santilli et al. Revisiting Uncertainty Quantification Evaluation in Language Models: Spurious Interactions with Response Length Bias Results. ACL 2025. \
> [4] Sriramanan et al. LLM-Check: Investigating Detection of Hallucinations in Large Language Models. NeurIPS 2024.  \
> [5] Vashurin et al. Benchmarking Uncertainty Quantification Methods for Large Language Models with LM-Polygraph. TACL 2025. \
> [6] Vashurin et al. UNCERTAINTY-LINE: Length-Invariant Estimation of Uncertainty for Large Language Models. EMNLP 2025 \
> [7] Vazhentsev et al. Token-Level Density-Based Uncertainty Quantification Methods for Eliciting Truthfulness of Large Language Models. NAACL 2025.  \
> [8] Vazhentsev et al. Unconditional Truthfulness: Learning Conditional Dependency for Uncertainty Quantification of Large Language Models. EMNLP 2025 \
> [9] Zhang et al. Enhancing Uncertainty-Based Hallucination Detection with Stronger Focus. EMNLP 2023.

---

> ### Author Response · Authors · 2025-11-26
>
> Dear Reviewer,
>
> Thank you again for your time and thoughtful feedback. With one week remaining in the discussion period, we would greatly appreciate it if you could consider our responses. Our rebuttal addresses the concerns raised in your review. If there are no remaining concerns, we would appreciate it if you could reconsider your scores accordingly. Should any questions remain, we would be happy to provide further clarification and engage in further discussion.

---

### Official Review · Reviewer_qQ9q · 2025-10-31

**Soundness:** 3
**Presentation:** 3
**Contribution:** 3
**Rating:** 6
**Confidence:** 3

**Summary:**

The paper proposes RAUQ (Recurrent Attention-based Uncertainty Quantification), a lightweight and unsupervised hallucination detection approach for large language models (LLMs).RAUQ identifies “uncertainty-aware attention heads” that display attention degradation when models produce hallucinated content. The method recursively propagates token-level uncertainty based on attention and token probabilities using only a single forward pass, incurring less than 1% extra computation. Extensive experiments across 4 LLMs (e.g., Llama-3.1, Gemma-2) and 12 tasks demonstrate consistent performance improvements over more than 15 baseline methods.

**Strengths:**

1.	RAUQ Leverages “uncertainty-aware” attention heads for hallucination detection is a creative and intuitive direction that remains computationally efficient and easy to implement in white-box settings. The authors evaluate multiple models and tasks, showing solid gains in PRR and selective generation metrics, with thorough ablation studies on α, layer ranges, and head selection.
2.	RAUQ operates in a single forward pass and introduces negligible computational overhead (<1%), which makes it suitable for real-time or large-scale deployment.
3.	The work provides both quantitative and qualitative insights into why certain attention heads correlate with uncertainty, offering interpretability beyond pure calibration metrics.

**Weaknesses:**

1. While empirical findings are convincing, the paper lacks a rigorous theoretical explanation of why specific heads consistently reflect uncertainty. A deeper causal or functional analysis would enhance scientific understanding.
2. RAUQ requires access to internal attention weights, limiting its applicability in black-box LLMs such as API-based commercial models. Discussion on possible extensions or alternative signals for black-box use would strengthen the paper.

**Questions:**

1.	How are attention weights normalized and extracted from different architectures (e.g., Llama vs Gemma) to ensure consistent computation of RAUQ? How stable the identified “uncertainty-aware heads” remain across different prompt styles or lengths?
2.	Lines 270–289 — the pseudocode formatting in Algorithm 1 should be adjusted. The current line numbers exceed the right boundary, and the table layout contains excessive white space, which negatively affects readability.

---

> ### Author Response · Authors · 2025-11-21
>
> Thank you very much for your time in reviewing our paper and your very valuable comments.
>
> **Q1. While empirical findings are convincing, the paper lacks a rigorous theoretical explanation of why specific heads consistently reflect uncertainty.**
>
> A1. Our method integrates three key ideas:
> 1. Hallucination detection requires uncertainty propagation from previous generation steps to overcome the problem of their conditional dependency.
> 2. Attention weights to previous tokens contain patterns indicative for hallucination detection.
> 3. Selecting the most contrastive attention head is important for high performance.
>
> The first two ideas are based on the theoretical background covered in (Sriramanan et al. 2024) and (Vazhentsev et al. 2025). The last idea that only specific attention heads provide valuable information for hallucination detection is a heuristic but is well supported by prior mechanistic interpretability studies of attention heads, which have shown that different heads serve distinct functions (Elhelo et al. 2025). Thus, our empirical finding is consistent with previous research.
>
> Moreover, incorporating this RAUQ component into the recently proposed AttentionScore method (Sriramanan et al., 2024) substantially improves its performance. While Sriramanan et al. use attention weights, motivating it through an eigen-analysis of attention kernels, they do not perform attention-head selection. Experiment presented in Section 5.3 (lines 493-505) clearly shows that introducing our head-selection component yields a significant performance boost not just for RAUQ, but also for AttentionScore.
>
> To address your concern, we have added a subsection in Section 4 that provides a broader discussion of the theoretical grounding of the method’s components and connection to other methods based on attention and mechanistic analysis.
>
> **Q2. How stable the identified “uncertainty-aware heads” remain across different prompt styles or lengths?**
>
> A2. Overall, while minor variations exist, the same heads generally provide the most informative signals for RAUQ, supporting both intra- and cross-task stability across languages, prompt styles, and answer lengths.
>
> The stability of the selected heads is investigated in Tables 9 and 10 in Appendix C.2, which present the chosen heads for WMT14 De-En and CoQA. First, the results show that in most layers, the most frequently selected head appears in over 90% of cases, indicating strong stability. Even in less consistent layers, the top-3 heads still cover more than 90% of instances. Second, the selected heads also align well across datasets, demonstrating cross-task consistency.
>
> **Q3. RAUQ requires access to internal attention weights, limiting its applicability in black-box LLMs such as API-based commercial models. Discussion on possible extensions or alternative signals for black-box use would strengthen the paper.**
>
> A3. In this work, we focus on white-box UQ methods -- a group of techniques that assume full access to the internal states of an LLM. Although such methods cannot be directly applied to black-box models (LLMs exposed only through API), our work illustrates that white-box access enables substantial performance improvements while keeping the method highly computationally efficient. Consequently, our approach paves the way for integrating robust UQ mechanisms directly into existing LLM-as-a-service systems, which is highly useful for real-world applications.
>
> Nevertheless, one possible way of adapting our technique to a black-box setting is to employ an auxiliary white-box proxy LLM, from which attention signals and logits can be extracted. Applying such a proxy model can be effective because it allows detecting ambiguous or underspecified queries, thereby capturing uncertainty patterns that partially mirror those of the black-box target model.
>
> We have expanded the discussion on the black-box applicability of RAUQ in the Limitations section of the new version of the paper.
>
> **Q4. How are attention weights normalized and extracted from different architectures (e.g., Llama vs Gemma) to ensure consistent computation of RAUQ?**
>
> A4. We implement a model-specific normalization procedure that converts raw attention weights into a lower-triangular attention matrix in a uniform manner across all architectures. We carefully verified the correctness of this transformation, and the exact implementation is available in the supplementary code attached to the submission.
>
> To address your concern, we have added a discussion in Section 5.1 on lines 354-356.
>
> **Q5. Lines 270–289 — the pseudocode formatting in Algorithm 1 should be adjusted. The current line numbers exceed the right boundary, and the table layout contains excessive white space, which negatively affects readability.**
>
> A5. Thank you for pointing this out! We have fixed the pseudocode formatting in Algorithm 1.

---

> > ### Author Response · Authors · 2025-11-21
> >
> > **References:**
> >
> > [1] Aichberger et al. Rethinking Uncertainty Estimation in Natural Language Generation. NeurIPS 2024.
> > [2] Elhelo et al. Inferring Functionality of Attention Heads from their Parameters. ACL 2025.
> > [3] Sriramanan et al. LLM-Check: Investigating Detection of Hallucinations in Large Language Models. NeurIPS 2024.
> > [4] Vazhentsev et al. Unconditional Truthfulness: Learning Conditional Dependency for Uncertainty Quantification of Large Language Models. EMNLP 2025.

---

> ### Author Response · Authors · 2025-11-26
>
> Dear Reviewer,
>
> Thank you again for your time and thoughtful feedback. With one week remaining in the discussion period, we would greatly appreciate it if you could consider our responses. Our rebuttal addresses the concerns raised in your review. If there are no remaining concerns, we would appreciate it if you could reconsider your scores accordingly. Should any questions remain, we would be happy to provide further clarification and engage in further discussion.

---

### Official Review · Reviewer_WU7n · 2025-10-31

**Soundness:** 3
**Presentation:** 3
**Contribution:** 3
**Rating:** 6
**Confidence:** 4

**Summary:**

This work propose Recurrent Attention-based Uncertainty Quantification (RAUQ), an unsupervised and efficient framework for identifying hallucinations. RAUQ leverages the behavior of “uncertainty-aware” attention heads to further improve the Focus approach by Zhang et al., 2023. Experiments on question answering, summarization, and translation tasks show that RAUQ consistently outperforms SOTA baselines. The low resource cost make RAUQ plug-and-play for real-time hallucination detection in white-box LLMs.

**Strengths:**

1. The analysis of uncertainty-aware attention heads is both interesting and sound, lending interpretability to the subsequent improvements made to the Focus approach.

2. Experiments on 12 relevant benchmarks demonstrate the consistent superiority of the proposed approach. Further analysis of the framework modules and computational costs reinforces the strength of this work.

**Weaknesses:**

1. The experiments only conduct on UQ methods, other hallucination detection works based on external information can also be introduced.

2. This work is based on small-scale LLMs with approximately 8B parameters. Future experiments on larger models (30B or 70B parameters) are expected to yield improved soundness.

**Questions:**

1. What does "Simple Focus" mean? There is no description about this term. Do you mean a simplified version of the original Focus Approach by Zhang et al., 2023?

---

> ### Author Response · Authors · 2025-11-21
>
> Thank you very much for your time in reviewing our paper and your insightful comments.
>
> **Q1. This work is based on small-scale LLMs with approximately 8B parameters. Future experiments on larger models (30B or 70B parameters) are expected to yield improved soundness.**
>
> A1. Because architectures and pre-training procedures are largely consistent across model sizes, RAUQ generalizes effectively to both larger and smaller LLMs. To demonstrate this, we have conducted additional experiments using SmolLM2-360M, Llama-3.2-1B, and LLaMA-3.1-70B. For models ≤1B parameters, we exclude MMLU, GSM8K, and MedQUAD due to their near-zero performance on these tasks. Below we provide a table comparing RAUQ with the most prominent baselines identified in Table 1.
>
> The results show that RAUQ is the best method for QA and MT on ≤1B LLMs, and for Summ and MT on 70B LLM. Overall, RAUQ surpasses the second-best method by an average of 4% of PRR across all tasks and models. These results highlight the strong generalization ability of RAUQ across a wide range of model sizes.
>
> We have included this experiment in the new version of the paper in Table 12 in Appendix D.1.
>
> | Methods | SmolLM2-360M -- QA | SmolLM2-360M -- Summ | SmolLM2-360M -- MT | Llama-3.2-1B -- QA | Llama-3.2-1B -- Summ | Llama-3.2-1B -- MT | Llama-3.1-70B -- QA | Llama-3.1-70B -- Summ | Llama-3.1-70B -- MT | Mean |
> |:--------|-------------------:|--------------------:|--------------------:|-------------------:|--------------------:|--------------------:|---------------------:|----------------------:|----------------------:|------:|
> | MSP               |                                   0.36 |                                    **0.45** |                                    0.33 |                             0.32 |                              **0.51** |                              0.35 |                                 0.36 |                                  0.13 |                                  0.45 |           0.36 |
> | Perplexity        |                                   0.37 |                                    0.33 |                                    0.49 |                             0.31 |                              0.39 |                              0.43 |                                 0.32 |                                  0.25 |                                  0.34 |           0.36 |
> | Attention Score   |                                   0.07 |                                    0    |                                    0.12 |                             0.05 |                              0.03 |                              0.1  |                                 0.05 |                                  0.05 |                                  0.21 |           0.08 |
> | SAR               |                                   0.34 |                                    0.09 |                                    0.35 |                             0.3  |                              0.09 |                              0.41 |                                 0.38 |                                  0.11 |                                  0.37 |           0.27 |
> | Semantic Density  |                                   0.36 |                                    0.21 |                                    0.26 |                             0.35 |                              0.22 |                              0.28 |                                 **0.39** |                                  0.1  |                                  0.24 |           0.27 |
> | RAUQ              |                                   **0.43** |                                    0.36 |                                   **0.49** |                             **0.36** |                              0.42 |                              **0.49** |                                 0.36 |                                  **0.25** |                                  **0.46** |           **0.4**  |
>
> **Q2. The experiments only conduct on UQ methods, other hallucination detection works based on external information can also be introduced.**
>
> A2. Methods based on external knowledge sources, such as search engines or auxiliary LLMs, come with their own limitations and advantages, placing them in a fundamentally different research direction that is not directly comparable to the uncertainty-based methods studied here. Uncertainty-based approaches assess the reliability of model outputs using only the model’s internal capabilities, which is essential in many practical scenarios. These methods are not competitors to external-knowledge approaches but are often complementary; for instance, UQ can be applied on top of RAG.
>
> Therefore, in this work, we focus exclusively on uncertainty-based techniques, and methods relying on external knowledge fall outside our scope.

---

> ### Author Response · Authors · 2025-11-21
>
> **Q3. What does "Simple Focus" mean? There is no description about this term. Do you mean a simplified version of the original Focus Approach by Zhang et al., 2023?**
>
> A3.  SimpleFocus is a simplified variant of the Focus method introduced by Zhang et al. (2023). It preserves only the core scoring components: attention-based signals and greedy log-likelihood (omitting a proxy model, IDF-based keywords, and NER). Despite its minimalism, it often produces better results than the original Focus method. In the paper, we report results for both variants. Thank you for noting the missing description; we have added it to the new version of the paper in Section 5.1 on lines 363-365.
>
> **References**
>
> [1] Zhang et al. Enhancing Uncertainty-Based Hallucination Detection with Stronger Focus. EMNLP 2023.

---

> ### Author Response · Authors · 2025-11-26
>
> Dear Reviewer,
>
> Thank you again for your time and thoughtful feedback. With one week remaining in the discussion period, we would greatly appreciate it if you could consider our responses. Our rebuttal addresses the concerns raised in your review. If there are no remaining concerns, we would appreciate it if you could reconsider your scores accordingly. Should any questions remain, we would be happy to provide further clarification and engage in further discussion.

---

### Official Review · Reviewer_hpVq · 2025-11-05

**Soundness:** 3
**Presentation:** 3
**Contribution:** 3
**Rating:** 4
**Confidence:** 4

**Summary:**

The authors propose Recurrent Attention-based Uncertainty Quantification (RAUQ), a method that detects "uncertainty-aware" attention heads to identify hallucinations in NLG.

**Strengths:**

- The paper addresses a well-defined problem with a clear mechanistic insight: the link between attention head behavior and uncertainty.
- It is clearly written, well-structured, and the visualizations effectively support the main findings.
- RAUQ is fully unsupervised and adds less than 1% computational overhead, making it practical for real-time use when model internals are accessible.
- The experimental evaluation is comprehensive, covering 12 datasets, 4 models, and 3 generation tasks. RAUQ consistently outperforms strong baselines, and the ablation studies are detailed and informative.

**Weaknesses:**

- The intuition behind “uncertainty-aware” attention heads is entirely empirical and lacks theoretical grounding. There is no evidence that these attention patterns cause hallucinations rather than simply correlate with them.
- The main conceptual issue is inconsistency between motivation and method. The paper claims certain heads are consistently linked to factuality, yet the algorithm reselects new heads for each sequence based on current attention magnitudes. This per-sequence selection contradicts the idea of stable “uncertainty-aware” heads and weakens the mechanistic motivation.
- Although the authors claim that RAUQ requires no hyperparameter tuning, the balancing coefficient $\alpha$ is tuned on a held-out set, and Figure 5 shows that performance can significantly change with its value. Hence, this claim is overstated.
- Comparisons focus mainly on attention-based or sampling baselines. Recent approaches using mechanistic interpretability or calibration-based uncertainty estimation are not included.
- Minor: The method requires access not only to token probabilities but also to internal attention weights, restricting its broader applicability.

**Questions:**

- How stable are the selected attention heads across different sequences or inputs? Do the same heads tend to be selected consistently, or does the selection vary substantially? How do the authors justify selecting different heads per input sequence?
- Which sampling temperature are the output sequences generated with? Table 3 indicates a temperature of 1.0. Howerver, Aichberger et al. [1] show that the greedy output sequence should be used for computing the MSP to obtain the highest performance. It would be interesting to compare against this baseline as well.

---

[1] Lukas Aichberger, Kajetan Schweighofer, and Sepp Hochreiter. Rethinking uncertainty estimation in natural language generation. arXiv preprint arXiv:2412.15176, 2024.

---

> ### Author Response · Authors · 2025-11-21
>
> Thank you very much for your time in reviewing our paper and your insightful comments.
>
> **Q1. The intuition behind “uncertainty-aware” attention heads is entirely empirical and lacks theoretical grounding.**
>
> A1. The theoretical motivation of RAUQ builds on prior attention-based UQ methods, such as AttentionScore (LLM-Check) (Sriramanan et al., NeurIPS 2024), StrongerFocus (Zhang et al., 2023), and TAD (Vazhentsev et al, 2025).
>
> Sriramanan et al. (2024) illustrate that attention weights contain patterns indicative of hallucinations through eigen-analysis of attention kernels. They justify using only the attention weights to the previous token, as these correspond to the eigenvalues of the lower triangular attention matrix, and their sum exactly equals its log determinant. In our work, we reveal a similar pattern through a mechanistic analysis of attention weights, examining the correlation between hallucinations and attention weight distributions.
>
> Zhang et al. (2023) and Vazhentsev et al. (2025) illustrate that spotting LLM hallucinations require recurrent uncertainty propagation from previous generation steps as probability distribution modeled by the LLM is a conditional distribution. We directly leverage this theoretical finding in the recurrent aggregation formula (2).
>
> Our main heuristic: attention head selection is based on our observation that the majority of heads are not indicative of hallucinations (Figure 3a). Our approach simply selects the most contrastive head that has the best potential for discriminating between hallucinations and non-hallucinations.
>
> Thus, RAUQ is well-motivated as it builds on theoretical findings in previous work. Moreover, our insights helped to improve the previous attention-based method AttentionScore. Section 5.3, lines 493-505, shows that integrating RAUQ components into AttentionScore (Sriramanan et al., 2024) significantly boosts its performance.
>
> To address your concern, we have added a subsection in Section 4 that provides a broader discussion of the theoretical grounding of the method’s components and connection to other methods based on attention and mechanistic analysis.
>
> **Q2. The main conceptual issue is inconsistency between motivation and method. The paper claims certain heads are consistently linked to factuality, yet the algorithm reselects new heads for each sequence based on current attention magnitudes.**
>
> A2. The optimal attention heads exhibit strong stability within each task (see Tables 9 and 10 in Appendix C.2). Moreover, when comparing the selected heads across the two datasets, we observe substantial overlap. In several layers (e.g., 10, 12, 13, 15, 16, and 20) the selected attention heads align perfectly, providing clear evidence of strong cross-task consistency and reinforcing our motivation behind uncertainty-aware attention heads.
>
> Nevertheless, retaining dynamic, unsupervised head selection as part of the algorithm fully removes the need for any precise task-specific adjustments. This design choice ensures that the method remains entirely unsupervised, requires no validation data, and is seamlessly plug-and-play for any new LLM or task.
>
> To further demonstrate the effectiveness of our strategy, we conduct an additional ablation in which a single optimal head per layer is selected for all inputs determined on a small held-out validation set (see table below). The results show that the gains from such precise per-dataset head selection are marginal, and the average performance across all datasets remains effectively similar. This indicates that our dynamic, fully unsupervised strategy already achieves near-optimal performance. We have included this ablation study in the new version of the paper in Section 5.3 and Table 11 in Appendix C.2.
>
> | Method   |   XSUM | SamSum | CNN | WMT14 |   WMT19 |  TruthfulQA | CoQA |   SciQ |   TriviaQA | MMLU | GSM8k |   Mean |
> |:----------------------|---------------------:|---------------------:|---------------------:|---------------------:|---------------------:|---------------------:|---------------------:|---------------------:|-------------------------:|-------------------------:|-----------------------------:|---------------:|
> | RAUQ                 |                    **0.384** |                      0.423 |                   0.189 |                0.406 |                **0.488** |                          **0.399** |                    0.248 |                    **0.506** |                        **0.548** |                  0.513 |                   0.323 |          **0.402** |
> | RAUQ (Single Head)   |                    0.382 |                      **0.426** |                   **0.195** |                **0.407** |                0.481 |                          0.386 |                    **0.257** |                    0.494 |                        0.544 |                  **0.528** |                   **0.325** |          **0.402** |

---

> ### Author Response · Authors · 2025-11-21
>
> **Q3. Although the authors claim that RAUQ requires no hyperparameter tuning, the balancing coefficient is tuned on a held-out set, and Figure 5 shows that performance can significantly change with its value. Hence, this claim is overstated.**
>
> A3. The method is very robust to the choice of $\alpha$. We tune $\alpha$ only on one task and one LLM and use the same value across all 12 datasets, 3 task types, and 4 LLMs.
> All Table 1 results use fixed $\alpha=0.2$ and consistently outperform the baselines.
>
> Figure 5 shows that for each task, it is possible to select an “optimal” alpha if we specifically tune it on its validation set, but these almost always fall between 0.1 and 0.3 -- very close to our choice. Therefore, RAUQ can be used in a plug-and-play fashion without task- or model- specific tuning of $\alpha$.
>
> **Q4. Recent approaches using mechanistic interpretability or calibration-based uncertainty estimation are not included.**
>
> A4. To the best of our knowledge, we cover all relevant state-of-the-art UQ methods. The set of baselines includes more than 20 baselines, encompassing all known information- and sampling-based baselines, methods based on mechanistic interpretability that analyze attention patterns and hidden states (Zhang et al., 2023; Sriramanan et al., 2024), as well as supervised hallucination detection approaches (Tables 12 and 13 in Appendix).
>
> We would be grateful if the reviewer could provide specific suggestions for additional baselines. We would be happy to incorporate them to further strengthen our experimental evaluation.
>
> **Q5. Minor: The method requires access not only to token probabilities but also to internal attention weights, restricting its broader applicability.**
>
> A5. In this work, we focus on white-box UQ methods -- techniques that assume full access to the internal states of an LLM. Although as we mention in the Limitation section, such methods cannot be directly applied to black-box models (e.g. LLMs exposed only through API), our work demonstrates that white-box access enables substantially performance improvements, while remaining computationally efficient. Consequently, our approach paves the way for integrating robust UQ mechanisms directly into existing LLM-as-a-service systems, which is highly useful for real-world applications.
>
> Nevertheless, one possible direction for adapting our technique to a black-box setting is to employ an auxiliary white-box proxy LLM from which attention signals and logits can be extracted. Such a proxy model may be effective because it can detect ambiguous or underspecified queries, thereby capturing uncertainty patterns that partially mirror those of the black-box target model.
>
> **Q6. How stable are the selected attention heads across different sequences or inputs? Do the same heads tend to be selected consistently, or does the selection vary substantially?**
>
> A6. We examine the stability of head selection in Tables 9 and 10 in Appendix C.2, which present the selected heads for WMT14 De-En and CoQA. First, the tables show that in most layers, the most frequently selected head appears in over 90% of cases, indicating strong stability. Even in less consistent layers, the top-3 heads still cover more than 90% of instances. Second, the selected heads also align well across datasets, indicating cross-task consistency.
>
> Overall, while minor variations exist, the same heads generally provide the most informative signals for RAUQ, supporting both intra- and cross-task stability.
>
> **Q7. How do the authors justify selecting different heads per input sequence?**
>
> A7. While uncertainty-aware heads are consistent, retaining dynamic, unsupervised head selection as part of the algorithm fully removes the need for any precise task-specific adjustments. This design choice ensures that the method remains entirely unsupervised, requires no validation data, and is seamlessly plug-and-play for any new LLM or task (see also A2 for additional ablation study).
>
> **Q8. Which sampling temperature are the output sequences generated with? … Aichberger et al. show that the greedy output sequence should be used for computing the MSP to obtain the highest performance.**
>
> A8. As shown in Table 3, we use greedy decoding to generate the main sequence, for which we compute uncertainty. Sampling is used solely to obtain multiple outputs for sampling-based baselines. Accordingly, the MSP score is always computed on the greedy output sequence, fully consistent with the setup in Aichberger et al. (G-NLL). We have clarified this in the new version of the paper in Appendix A.
>
> **References:**
>
> [1] Sriramanan et al. LLM-Check: Investigating Detection of Hallucinations in Large Language Models. NeurIPS 2024. \
> [2] Vazhentsev et al. Unconditional Truthfulness: Learning Conditional Dependency for Uncertainty Quantification of Large Language Models. EMNLP 2025. \
> [3] Zhang et al. Enhancing Uncertainty-Based Hallucination Detection with Stronger Focus. EMNLP 2023.

---

> ### Author Response · Authors · 2025-11-26
>
> Dear Reviewer,
>
> Thank you again for your time and thoughtful feedback. With one week remaining in the discussion period, we would greatly appreciate it if you could consider our responses. Our rebuttal addresses the concerns raised in your review. If there are no remaining concerns, we would appreciate it if you could reconsider your scores accordingly. Should any questions remain, we would be happy to provide further clarification and engage in further discussion.

---

### Author Response · Authors · 2025-11-24
**Summary of Revisions**

Dear Reviewers and Area Chair,

Thank you for taking the time to review and evaluate our work. We greatly appreciate your efforts and valuable feedback. Below is a summary of the **main changes** we have made in the revised version of the paper to address the reviewers’ feedback:

1. Added a theoretical subsection at the beginning of Section 4, explaining the rationale for the method’s key components: the grounding in uncertainty-aware heads, the appropriateness of recurrence, and why the components work together (requested by hpVq, qQ9q, and KfLE).
2. Enhanced the explanation in Section 4 regarding key design choices, including token-level and layer-wise aggregation, the selected layer range, and the criteria for attention head selection (requested by KfLE). Clarified the choice of $\alpha$ and demonstrated the robustness of using a fixed hyperparameter value across different models and tasks (requested by hpVq and KfLE).
3. Added experiments with LLMs of diverse sizes: SmolLM-2 360M, LLaMA-3.2 1B, and LLaMA-3.1 70B, in Table 12 in Appendix D.1 (requested by WU7n and KfLE).
4. Added an experiment using Layer Integrated Gradients (LIG), computed on the output projection layer, in place of attention scores in Section 5.3 and Table 16 in Appendix D.4. This extension demonstrates the method’s applicability to models with non-standard or without attention mechanisms (requested by KfLE).
5. Added a new ablation study in which a single optimal attention head per layer is selected on a small validation set. Results are presented in Section 5.3 and Table 11 in Appendix C.2 (requested by hpVq).
6. Added a qualitative analysis at the end of Section 5.3, including an error-type analysis and a discussion of confident hallucinations (requested by KfLE).
7. Clarified the MSP computation for the greedy-decoded main sequence in Appendix A (requested by hpVq).
8. Expanded discussion of the white-box nature of the RAUQ method and its potential black-box applicability through the proxy-model extension in the Limitations section (requested by hpVq, qQ9q, and KfLE).
9. Added a clear definition of the SimpleFocus method in Section 5.1 (requested by WU7n).
10. Clarified attention extraction and normalization across different architectures in Section 5.1 (requested by qQ9q).
11. Adjusted the pseudocode formatting in Algorithm 1 to ensure line numbers do not exceed the right boundary (requested by qQ9q).

We would be happy to provide any further clarifications and updates if requested.

---

### Meta-Review · Area_Chair_ATAW · 2025-12-13

**Summary:**

The authors provide clarifications and additional experiments and analysis on the stability of uncertainty-aware heads, the justification for dynamic head selection, and the decoding setup. In addition, the authors provide experiments on external information.
These points are now supported and solved the reviewers' concerns. However, two issues remain only partially addressed. The theoretical basis for why specific attention heads capture uncertainty is still empirical, despite references to prior work, and the conceptual gap mentioned by the reviewer persists. In addition, the issue about missing calibration-based baselines is not completely resolved, as the rebuttal does not justify their exclusion. Finally, although the authors argue that the proposed model is robust to the $\alpha$, the original claim of “no tuning” remains somehow overstated.

**Reviewer Concerns:**

Reviewer hpVq comments are partially addressed. The authors supplement additional experiments that addressed some of the concerns, but, the comment regarding the theoretical basis that why attention heads capture uncertainty and the robustness to the hyperparameter is not convincing.
The concerns of reviewers WU7n and qQ9q are well addressed. Reviewer KfLE raised several weaknesses in the pre-rebuttal. The authors responded well to most of them. However, the authors did not provide the justification for Eq. 2 that shows the propagation of uncertainty.

**Reviewer Scores:**

Reviewers WU7n and qQ9q concerns are well addressed.
Reviewer hpVq and KfLE comments are not fully addressed.

---

### Decision · Program_Chairs · 2026-01-26

Reject